# Near-optimal Regret Using Policy Optimization
# in Online MDPs with Aggregate Bandit Feedback

Tal Lancewicki [1]  Yishay Mansour [1] [2]

## Abstract

We study online finite-horizon Markov Decision Processes with adversarially changing loss and aggregate bandit feedback (a.k.a full-bandit). Under this type of feedback, the agent observes only the total loss incurred over the entire trajectory, rather than the individual losses at each intermediate step within the trajectory. We introduce the first Policy Optimization algorithms for this setting. In the known-dynamics case, we achieve the first *optimal* regret bound of $\tilde{\Theta}(H^2\sqrt{SAK})$, where $K$ is the number of episodes, $H$ is the episode horizon, $S$ is the number of states, and $A$ is the number of actions. In the unknown dynamics case we establish regret bound of $\tilde{O}(H^3S\sqrt{AK})$, significantly improving the best known result by a factor of $H^2S^5A^2$.

## 1. Introduction

The standard model of reinforcement learning (RL) assumes a rich feedback loop, where for each step within the episode the agent observes the loss in that state as feedback. While ideal, this is often not the case in real-world applications. For example, in multi-turn dialogues with an LLM, feedback is typically available only at the end of the entire dialogue, not for each intermediate response. Similarly, in robotic manipulation, feedback is often only available for the entire trajectory, indicating whether the robot successfully completed its task, rather than providing feedback at every step of the robot's movement.

To address this challenge, Efroni et al. (2021) have initiated the study of aggregate bandit feedback in a stochastic setting where losses are generated in an i.i.d. manner. More recently, Cassel et al. (2024) have extended this to linear MDPs. Most related to our work is that of Cohen et al.

(2021b), who considered the setting we study here, where the losses are non-stochastic and may be chosen by an adversary. They introduce a variant of bandit linear optimization which they call Distorted Linear Bandits and provide a solution for it. Through the framework of occupancy measures, they reduce the problem of adversarial MDPs with aggregate bandit feedback to Distorted Linear Bandits. This approach allows them to obtain a regret of $\tilde{O}(H^5S^6A^{5/2}\sqrt{K})$, where $K$ is the number of episodes, $H$ is the horizon, $S$ is the number of states and $A$ is the number of actions. While the dependency on $K$ is optimal, the dependency on other parameters is far from optimal.

In this paper, we revise the setting considered in Cohen et al. (2021b), both in the known and unknown dynamics cases. We present algorithms based on the Policy Optimization framework (Cai et al., 2020; Shani et al., 2020; Luo et al., 2021; Chen et al., 2022a), which has strong connections to many practical algorithms such as NGP (Kakade, 2001), TRPO (Schulman et al., 2015), and PPO (Schulman et al., 2017). Our algorithms have a closed-form update and are more efficient than that of Cohen et al. (2021b), which requires solving a convex optimization problem in each iteration. We obtain the first optimal bound under known dynamics and significantly improve the regret bound of Cohen et al. (2021b) in the unknown dynamics case.

**Summary of Contributions.** The main contributions of the paper are as follows:

- We present the first Policy Optimization algorithms for Online MDPs with aggregate bandit feedback.
- Under known dynamics we are the first to establish the near-optimal regret bound of $\tilde{\Theta}(H^2\sqrt{SAK})$.
- In the unknown dynamics case, we achieve a regret of $\tilde{O}(H^3S\sqrt{AK})$. Surprisingly, this regret bound matches the best known regret for Policy Optimization with semi-bandit feedback (Luo et al., 2021).[1]
- We establish a new lower bound $\Omega(H^2\sqrt{SAK})$ for online MDPs with aggregate bandit feedback. To the

---

[1]Blavatnik School of Computer Science, Tel Aviv University, Israel [2]Google Research, Tel Aviv, Israel. Correspondence to: Tal Lancewicki <lancewicki@mail.tau.ac.il>.

*Proceedings of the $42^{nd}$ International Conference on Machine Learning*, Vancouver, Canada. PMLR 267, 2025. Copyright 2025 by the author(s).

[1]Luo et al. (2021) presents a slightly different dependence on the horizon $H$. This is due to their assumption of a loop-free MDP, which effectively enlarges their state space by a factor of $H$ compared to our model — see Remark 1.1.

best of our knowledge, this is the first lower bound for this setting that is not directly implied from the semi-bandit case.

A comparison of our results to previous works is summarized in Table 1.

**Overview of techniques.** Much of our algorithms' design and analysis follows the seminal work of Luo et al. (2021). As most of the regret minimization literature, they built upon the fundamental value difference lemma (Kakade & Langford, 2002; Even-Dar et al., 2009) that breaks the total regret as a weighted sum of local regrets in each state separately with respect to the $Q$-function. Our central observation is that the regret can be decomposed in a similar manner but with respect to different quantities, which we call the $U$-values. These quantities are particularly natural in the context of aggregate bandit feedback. The $U$-function at a given state and action is the expected cost **on the entire trajectory** given that we visit this state-action pair. Our decomposition with respect to the $U$-function is especially useful in the setting of aggregate bandit feedback since the $U$-function can be easily estimated using only the accumulated trajectory loss. This is in contrast to $Q$-function estimation that typically uses individual losses or the cost-to-go from a state-action pair.

### 1.1. Related work

**Aggregate bandit feedback with stochastic i.i.d losses** was first studied by Efroni et al. (2021) who obtained regret of $\tilde{O}(H^{3/2}S^2A^{3/2}\sqrt{K})$ with an efficient algorithm. Their transition and loss functions are not horizon-dependent. Adapting their bound for horizon-dependent losses and transitions as we consider here would effectively inflate the number of states by a factor of $H$, resulting in a regret bound of $\tilde{O}(H^{7/2}S^2A^{3/2}\sqrt{K})$. Cassel et al. (2024) introduced the first algorithm for Linear MDPs with stochastic losses and aggregate bandit feedback. Their algorithm attains regret of $\tilde{O}(\sqrt{d^5H^7K})$ where $d$ is the dimension of the feature map. In the special case of tabular MDPs, they show a regret of $\tilde{O}(H^{7/2}S^2A^{3/2}\sqrt{K})$.

**Adversarial Linear bandits** (see for example Lattimore & Szepesvári (2020)) is a variant of the classical Multi-armed Bandit problem where each action is associated with a vector in $\mathbb{R}^d$. The loss in each round is the inner product of the action with an unknown parameter vector chosen by an adversary. Through the concept of *occupancy measures*, online MDPs with aggregate bandit feedback and known dynamics can be seen as a special case of Adversarial Linear bandits. In terms of regret bounds, EXP2 (Dani et al., 2007; Cesa-Bianchi & Lugosi, 2012) with a specific exploration distribution achieves the optimal bound of $\Theta(B\sqrt{dK\log N})$ for any finite set of $N$ actions in $\mathbb{R}^d$,

where $B$ is a bound on the losses (Bubeck et al., 2012). Using a discretization argument, this bound can be extended to $\tilde{O}(Bd\sqrt{K})$ for any compact convex set. However, the EXP2 algorithm is not efficient in general. The latter bound for general convex set is attainable with efficient algorithms (polynomial in $d$) under mild assumptions (Hazan & Karnin, 2016). When using occupancy measures to reduce online MDPs with aggregate bandit feedback to Linear bandits, the decision set is of dimension $d = HSA$, the bound of the loss in each round is $B = H$ and the number of deterministic policies is $N = A^{SH}$. This results in regret of $\tilde{O}(H^2S\sqrt{AK})$ and $\tilde{O}(H^2SA\sqrt{K})$ with inefficient and efficient algorithms, respectively. For the known dynamics case, we improve these bounds to optimal $\tilde{\Theta}(H^2\sqrt{SAK})$ regret with an efficient and more natural algorithm.

As mentioned before, Cohen et al. (2021b) have extended the linear bandit model to *Distorted Bandit Online Linear Optimization* (DBOLO). They show that online MDPs with aggregate bandit feedback and *unknown* dynamics can be reduced to DBOLO efficiently. Their algorithm is built upon the SCRIBLE algorithm (Abernethy et al., 2008) and guarantees regret of $\tilde{O}(H^5S^6A^{5/2}\sqrt{K})$. On the same setting, we improve their regret bound to $\tilde{O}(H^3S\sqrt{AK})$.

**Regret minimization in MDPs with semi-bandit feedback** is extensively studied in the literature, initiated with the seminal UCRL algorithm (Jaksch et al., 2010) for stochastic losses. Their model was later extended to the more general Online (adversarial) MDPs where the loss functions are arbitrarily chosen by an adversary. Most algorithms for this model are based either on the framework of occupancy measures (Zimin & Neu, 2013; Rosenberg & Mansour, 2019a;b; Jin et al., 2020) or the Policy Optimization framework (Even-Dar et al., 2009; Shani et al., 2020; Luo et al., 2021). In the adversarial model with semi-bandit feedback and known dynamics, the optimal regret bound is $\tilde{\Theta}(H\sqrt{SAK})$ case and is attained by an occupancy-measure based algorithm (Zimin & Neu, 2013). With PO, the state-of-the-art regret under known dynamics is $\tilde{O}(H^2\sqrt{SAK})$. We achieve the same bound with aggregate bandit feedback which in this case is optimal. Under unknown dynamics, the best known bound is $\tilde{O}(H^2S\sqrt{AK})$ and is also attained by an occupancy-measure based algorithm (Jin et al., 2020), while the best known lower bound is $\Omega(H^{3/2}\sqrt{SAK})$ (Jin et al., 2018). With policy optimization algorithm, the best known bound is $\tilde{O}(H^3S\sqrt{AK})$ (Luo et al., 2021). Although we are in a setting with less informative feedback, we match the latter bound.

*Remark* 1.1. We note that some of the literature on semi-bandit feedback, such as Jin et al. (2020); Luo et al. (2021), assumes *loop-free* MDPs. Under this assumption the state space consists of $H$ disjoint sets $\mathcal{S} = \mathcal{S}_1 \cup \mathcal{S}_2 \cup \cdots \cup \mathcal{S}_H$ such that in step $h$ the agent can only be found in states from the set $\mathcal{S}_h$. Effectively, this means that their state space

*Table 1.* Comparison of regret bounds for online MDPs with aggregate bandit feedback. In the 'Loss' column, 'adv.' denotes adversarial (non-stochastic) losses, while 'stoc.' denotes stochastic i.i.d losses. The regret bounds presented in this table ignore logarithmic and low-order terms.

| Algorithm | Dynamics | Loss | Regret |
|---|---|---|---|
| Reduction to Efficient Linear Bandits algorithm | known | adv. | $\sqrt{H^4 S^2 A^2 K}$ |
| Algorithm 1 (**ours**) | known | adv. | $\sqrt{H^4 SAK}$ |
| Lower bound | known | adv. | $\sqrt{H^4 SAK}$ |
| UCBVI-TS (Efroni et al., 2021) | unknown | stoc. | $\sqrt{H^7 S^4 A^3 K}$ [†] |
| REPO for tabular MDPs (Cassel et al., 2024) | unknown | stoc. | $\sqrt{H^7 S^4 A^3 K}$ |
| Reduction to DBOLO (Cohen et al., 2021b) | unknown | adv. | $\sqrt{H^{10} S^{12} A^5 K}$ |
| Algorithm 2 (**ours**) | unknown | adv. | $\sqrt{H^6 S^2 AK}$ |

is larger than ours by a factor of $H$. So for example the regret bound $\tilde{O}(H^2 S\sqrt{AK})$ in Luo et al. (2021) implies a bound of $\tilde{O}(H^3 S\sqrt{AK})$ in the transition model presented in this paper. We emphasize that these differences are rather artificial and not due to an actual difference in the regret.

**Stochastic binary trajectory feedback** was studied in Chatterji et al. (2021). In their model, the rewards are drawn from a logistic model that depends on features of the trajectory.

**Preference-based RL (PbRL)** is a model where the feedback is given in terms of preferences over a trajectory pair instead of rewards. This is partially related to our model in motivation. A partial list of works on PbRL includes (Saha et al., 2023; Chen et al., 2022b; Wu & Sun, 2023). For additional related work on the topic, see the above references.

## 2. Preliminaries

A finite-horizon episodic adversarial MDP is defined by a tuple $\mathcal{M} = (\mathcal{S}, \mathcal{A}, H, s_{init}, p, \{\ell^k\}_{k=1}^K)$, where $\mathcal{S}$ and $\mathcal{A}$ are state and action spaces of sizes $|\mathcal{S}| = S$ and $|\mathcal{A}| = A$, respectively, $H$ is the horizon, $s_{init}$ is the initial state and $K$ is the number of episodes. $p : \mathcal{S} \times \mathcal{A} \times [H] \to \Delta_{\mathcal{S}}$ is the *transition function* such that $p_h(s'|s, a)$ is the probability to move to $s'$ when taking action $a$ in state $s$ at time $h$. $\{\ell^k : \mathcal{S} \times \mathcal{A} \times [H] \to [0, 1]\}_{k=1}^K$ are *cost functions* chosen by an *oblivious adversary*, where $\ell_h^k(s, a)$ is the cost for taking action $a$ at $(s, h)$ in episode $k$.

A *policy* $\pi : \mathcal{S} \times [H] \to \Delta_{\mathcal{A}}$ is a function that gives the probability $\pi_h(a|s)$ to take action $a$ when visiting state

$s$ at time $h$. The value $V_h^\pi(s; \ell)$ is the expected cost of $\pi$ with respect to cost function $\ell$ starting from $s$ in time $h$, i.e., $V_h^\pi(s; \ell) = \mathbb{E}\left[\sum_{h'=h}^H \ell_{h'}(s_{h'}, a_{h'})|\pi, s_h = s\right]$, where the expectation is with respect to policy $\pi$ and transition function $p$, that is, $a_{h'} \sim \pi_{h'}(\cdot|s_{h'})$ and $s_{h'+1} \sim p_{h'}(\cdot|s_{h'}, a_{h'})$. The *Q-function* is defined by $Q_h^\pi(s, a; \ell) = \mathbb{E}\left[\sum_{h'=h}^H \ell_{h'}(s_{h'}, a_{h'})|\pi, s_h = s, a_h = a\right]$. The occupancy measure $\mu_h^\pi(s, a) = \Pr[s_h = s, a_h = a|\pi, s_1 = s_{init}]$ is the distribution that policy $\pi$ induces over state-action pairs in step $h$, and we denote $\mu_h^\pi(s) = \sum_{a \in \mathcal{A}} \mu_h^\pi(s, a)$.

**Learning protocol and Feedback**   The learner interacts with the environment for $K$ episodes. At the beginning of episode $k$, it picks a policy $\pi^k$, and starts in an initial state $s_1^k = s_{init}$. In each time $h \in [H]$, it observes the current state $s_h^k$, draws an action from the policy $a_h^k \sim \pi_h^k(\cdot|s_h^k)$ and transitions to the next state $s_{h+1}^k \sim p_h(\cdot|s_h^k, a_h^k)$. There are three types of loss feedback that are common in the literature:

- In *full-information* feedback, at the end of episode $k$ the agent observes the full cost function $\ell^k \in [0, 1]^{HSA}$.
- Under *bandit* feedback (a.k.a semi-bandit), the agent observes the loss function over the agent's trajectory, $\{\ell_h^k(s_h^k, a_h^k)\}_{h=1}^H$.
- Under *aggregate bandit* feedback (a.k.a full-bandit), the agent observes only the entire episode loss, $L_{1:H}^k = \sum_{h=1}^H \ell_h^k(s_h^k, a_h^k)$.

In all three settings, the trajectory $\{s_h^k, a_h^k\}_{h=1}^H$ is assumed to be fully observed. In this work, we assume aggregate bandit feedback, which is the least informative among the three above.

The goal of the learner is to minimize the *regret*, defined as the difference between the learner's cumulative expected cost and the best fixed policy in hindsight:

$$R_K = \sum_{k=1}^K V_1^{\pi^k}(s_{init}; \ell^k) - \sum_{k=1}^K V_1^{\pi^\star}(s_{init}; \ell^k),$$

where $\pi^\star = \arg\min_\pi \sum_{k=1}^K V_1^\pi(s_{init}; \ell^k)$ is the best fixed policy in hindsight.

**Value difference lemma.**   The analysis of policy optimization algorithms is often built upon a fundamental regret decomposition that follows the following lemma:

**Lemma 2.1** (Value Difference Lemma (Kakade & Langford, 2002; Even-Dar et al., 2009))**.** *For any loss function $\ell$ and any policies $\pi$ and $\pi'$,*

$$V_1^\pi(s_{init}; \ell) - V_1^{\pi'}(s_{init}; \ell)$$
$$= \sum_{h,s} \mu_h^{\pi'}(s) \langle \pi_h(\cdot \mid s) - \pi_h'(\cdot \mid s), Q_h^\pi(s, \cdot; \ell) \rangle. \quad (1)$$

---

[†]This regret bound is adapted to horizon-dependent transition and losses - see the related work section for more details.

*where $\langle \cdot, \cdot \rangle$ is the inner product.*

As a direct consequence we can break the regret as,

$$R_K = \sum_{h,s} \mu_h^{\pi^\star}(s) \sum_{k=1}^{K} \langle \pi_h^k(\cdot \mid s) - \pi_h^\star(\cdot \mid s), Q_h^k(s, \cdot) \rangle,$$

where, for each $h$ and $s$, the internal sum over $k$ can be seen as regret of a Multi-armed bandit problem with respect to the loss vectors $Q_h^k(s, \cdot)$.

**Additional notations.** Episode indices appear as superscripts and in-episode steps as subscripts. The notations $\tilde{O}(\cdot)$ and $\lesssim$ hide poly-logarithmic factors including $\log(K/\delta)$ for confidence parameter $\delta$. $[n] = \{1, 2, \ldots, n\}$. The indicator of event $E$ is $\mathbb{I}\{E\}$ and we denote $\mathbb{I}_h^k(s, a) = \mathbb{I}\{s_h^k = s, a_h^k = s\}$. We use the notations $V_h^k(s), Q_h^k(s, a), \mu_h^k(s, a)$ when the policy and cost are $\pi^k$ and $\ell^k$, respectively. The expectation conditioned on the policy $\pi^k$ is denoted by $\mathbb{E}_k$.

## 3. The $U$-function

Policy Optimization algorithms build upon the Value difference lemma, but since the $Q$-function is unknown (due to the bandit feedback), one would need to estimate it. The state-of-the-art PO algorithm for semi-bandit case (Luo et al., 2021) estimates the $Q$-function via importance sampling:

$$\hat{Q}_h^\pi(s, a; \ell) = \frac{\mathbb{I}\{s_h = s, a_h = s\} L_{h:H}}{\mu_h^\pi(s, a)}$$

where $L_{h:H} = \sum_{h'=h}^{H} \ell_{h'}(s_{h'}, a_{h'})$ is the realized loss-to-go from time $h$. To be more precise, (Luo et al., 2021) add a small bias at the denominator to better control its variance and use an upper confidence bound of $\mu_h^\pi(s, a)$ whenever the dynamics is unknown. Indeed $\hat{Q}_h^\pi(s, a; \ell)$ is an unbiased estimate of $Q_h^\pi(s, a; \ell)$. Note that with aggregate bandit feedback $L_{h:H}$ cannot be computed and it becomes unclear how to directly estimate the $Q$-function. For this reason we introduce a new quantity which we call the $U$-function. While the $Q_h^\pi(s, a; \ell)$ is the expected cost-to-go from time $h$ given that we visit state $s$ perform action $a$ at that time; the $U$-function is the expected cost **on the entire trajectory** given that we visit $s$ and perform $a$ at time $h$. That is,

$$U_h^\pi(s, a; \ell) := \mathbb{E}\left[\sum_{h'=1}^{H} \ell_{h'}(s_{h'}, a_{h'}) \Big| \pi, s_h = s, a_h = a\right].$$

The following lemma shows that the difference $U_h^\pi(s, a) - Q_h^\pi(s, a)$ does not depend on $a$.

**Lemma 3.1.** *For any Markovian policy $\pi$, loss function $\ell$ and $(h, s, a) \in [H] \times \mathcal{A} \times \mathcal{S}$,*

$$U_h^\pi(s, a; \ell) - Q_h^\pi(s, a; \ell) = W_h^\pi(s; \ell)$$

*where $W_h^\pi(s; \ell) := \mathbb{E}[\sum_{h'=1}^{h-1} \ell_{h'}(s_{h'}, a_{h'}) | \pi, s_h = s]$.*

*Proof.* Due to the Markov property and the fact that $\pi$ is a Markovian policy, the trajectory up to time $h - 1$ does not depend on the action taken at time $h$. Thus,

$$U_h^\pi(s, a; \ell) - Q_h^\pi(s, a; \ell)$$
$$= \mathbb{E}\left[\sum_{h'=1}^{h-1} \ell_{h'}(s_{h'}, a_{h'}) \Big| \pi, s_h = s, a_h = a\right]$$
$$= \mathbb{E}\left[\sum_{h'=1}^{h-1} \ell_{h'}(s_{h'}, a_{h'}) \Big| \pi, s_h = s\right] = W_h^\pi(s; \ell).$$

$\square$

As a corollary of Lemma 3.1 we obtain the Value Difference Lemma with respect to the $U$-function.

**Corollary 3.2.** *For any loss function $\ell$ and any policies $\pi$ and $\pi'$,*

$$V_1^\pi(s_{init}; \ell) - V_1^{\pi'}(s_{init}; \ell)$$
$$= \sum_{h,s} \mu_h^{\pi'}(s) \langle \pi_h(\cdot \mid s) - \pi_h'(\cdot \mid s), U_h^\pi(s, \cdot; \ell) \rangle.$$

*Proof.* For each $h$ and $s$, $\sum_a \pi_h(a \mid s) = \sum_a \pi_h'(a \mid s) = 1$. Thus,

$$\langle \pi_h(\cdot \mid s) - \pi_h'(\cdot \mid s), U_h^\pi(s, \cdot; \ell) - Q_h^\pi(s, \cdot; \ell) \rangle$$
$$= \sum_{a \in \mathcal{A}} (\pi_h(a \mid s) - \pi_h'(a \mid s)) W_h^\pi(s; \ell) = 0$$

Adding the above for each $h$ and $s$ in the sum on the right-hand side of Equation (1) completes the proof. $\square$

As in the $Q$-function case, a direct consequence is that we can break the regret as,

$$R_K = \sum_{h,s} \mu_h^{\pi^\star}(s) \sum_{k=1}^{K} \langle \pi_h^k(\cdot \mid s) - \pi_h^\star(\cdot \mid s), U_h^k(s, \cdot) \rangle$$

(2)

where similarly to other notations, we slightly abuse notation and write $U_h^k(s, a) = U_h^{\pi^k}(s, a; \ell^k)$.

In the context of aggregate bandit feedback, the $U$-function is much more useful as it can be estimated using the loss of the entire trajectory. In particular, the following is an unbiased estimator of $U_h^\pi(s, a; \ell)$:

$$\hat{U}_h^\pi(s, a; \ell) = \frac{\mathbb{I}\{s_h = s, a_h = s\} L_{1:H}}{\mu_h^\pi(s, a)}$$

where $L_{1:H} = \sum_{h=1}^{H} \ell_h(s_h, a_h)$ is the aggregated feedback of the trajectory. Later in the algorithms, we will use an optimistic variant of this estimator.

# 4. Known Dynamics

Our algorithm is based on the regret decomposition in Equation (2). Each internal sum can be seen as regret of a bandit problem with loss vectors $U_h^k(s, \cdot)$. Thus, we run a Multiplicative Weight Update with respect to an estimate of the $U$-function that uses only the aggregated trajectory loss: $\pi_h^{k+1}(\cdot \mid s) \propto \exp(\eta(\hat{U}_h^k(s, \cdot) - B_h^k(s, \cdot)))$. Here $\eta$ is a learning rate, $\hat{U}^k$ is the estimate of the $U$-function and $B^k$ is some bonus function that we'll define later. The estimate of the $U$-function is defined by,

$$\hat{U}_h^k(s, a) = \frac{\mathbb{I}_h^k(s, a)}{\mu_h^k(s, a) + \gamma} L_{1:H}^k. \qquad (3)$$

where $\gamma = \Theta(1/\sqrt{K})$ used to reduce the variance of the estimator. We note that in the known dynamics case $\mu_h^k(s, a)$ can be easily computed using dynamic programming.

Note that the regret decomposition in Equation (2) is averaged with respect to the state occupancy of $\pi^\star$, while the second moment of the estimator is roughly scaled inversely with the state occupancy of $\pi^k$. In order to control this distribution mismatch, we reduce from our $U$-estimate a bonus term $B_h^k(s, a)$ similar to Luo et al. (2021). $B_h^k(s, a)$ is essentially a $Q$-function with respect to a known loss function $b^k$. In the known dynamics case we set $b_h^k(s) = \sum_{a \in \mathcal{A}} \frac{3\gamma H \pi_h^k(a|s)}{\mu_h^k(s)\pi_h^k(a|s) + \gamma}$, where the reason for this particular choice will become clearer later in the analysis. $B_h^k(s, a)$ is computed using standard Bellman equations (e.g., as in Lancewicki et al. (2023)), which is simpler and more intuitive than the dilated version of Luo et al. (2021).

**Theorem 4.1.** *Under known dynamics, running Algorithm 1 with $\eta = (H\sqrt{SAK} + H^2\sqrt{K})^{-1}$ and $\gamma = 2\eta H$, guarantees with probability $1 - \delta$,*

$$R_K \leq \tilde{O}(H^2\sqrt{SAK} + H^3\sqrt{K}).$$

If $SA \geq H^2$ (which is typically the case since for any practical application $S \gg H$), the first term dominates and our bound is optimal up to logarithmic terms. A straightforward reduction of our problem to linear bandits that uses the efficient algorithm of Hazan & Karnin (2016) guarantees expected regret of $\tilde{O}(H^2 SA\sqrt{K})$. Our algorithm improves that by a factor of $\sqrt{SA}$ and guarantees regret with high probability rather than only in expectation. In addition, our algorithm is more computationally efficient since it has a closed form update as opposed to the reduction that requires solving a convex optimization problem in each iteration.

Before we outline the proof of Theorem 4.1, let us first show that $\hat{U}_h^k(s, a)$ is a nearly unbiased estimator of $U_h^k(s, a)$.

---

**Algorithm 1** Policy Optimization with Aggregated Bandit Feedback and Known Transition Function

---

**Input:** state space $\mathcal{S}$, action space $\mathcal{A}$, horizon $H$, learning rate $\eta > 0$, exploration parameter $\gamma > 0$, confidence parameter $\delta > 0$.

**Initialization:** Set $\pi_h^1(a \mid s) = 1/A$ for every $(h, s, a) \in [H] \times \mathcal{S} \times \mathcal{A}$.

**for** $k = 1, 2, \ldots, K$ **do**

    Play episode $k$ with policy $\pi^k$ and observe aggregated bandit feedback $L_{1:H}^k = \sum_{h=1}^H \ell_h^k(s_h^k, a_h^k)$.

    $\hat{U}_h^k(s, a) = \frac{\mathbb{I}_h^k(s, a)}{\mu_h^k(s, a) + \gamma} L_{1:H}^k$

    # Bonus Computation

    Set $B_{H+1}^k(s, a) = 0$ for every $(s, a) \in \mathcal{S} \times \mathcal{A}$.

    **for** $h = H, H - 1, \ldots, 1$ **do**

        **for** $(s, a) \in \mathcal{S} \times \mathcal{A}$ **do**

            $b_h^k(s) = \sum_{a \in \mathcal{A}} \frac{3\gamma H \pi_h^k(a|s)}{\mu_h^k(s)\pi_h^k(a|s) + \gamma}$

            $B_h^k(s, a) = b_h^k(s)$
                $+ \sum_{s',a'} p_h(s' \mid s, a)\pi_{h+1}^k(a' \mid s')B_{h+1}^k(s', a')$

        **end for**

    **end for**

    # Policy Improvement

    For every $(s, a, h) \in \mathcal{S} \times \mathcal{A} \times [H]$:

$$\pi_h^{k+1}(a \mid s) = \frac{\pi_h^k(a \mid s)e^{-\eta(\hat{U}_h^k(s,a) - B_h^k(s,a))}}{\sum_{a'} \pi_h^k(a' \mid s)e^{-\eta(\hat{U}_h^k(s,a') - B_h^k(s,a'))}}.$$

**end for**

---

**Lemma 4.2.** *Under Algorithm 1, for any $h, s, a$ and $k$,*

$$\mathbb{E}_k\left[\hat{U}_h^k(s, a)\right] = \frac{\mu_h^k(s, a)}{\mu_h^k(s, a) + \gamma} U_h^k(s, a)$$

*Proof.* By definition $\Pr(\mathbb{I}_h^k(s, a) = 1 \mid \pi^k) = \mu_h^k(s, a)$. Using the law of total expectation and the fact that $\hat{U}_h^k(s, a) = 0$ whenever $\mathbb{I}_h^k(s, a) = 0$ we get,

$$\mathbb{E}_k\left[\hat{U}_h^k(s, a)\right] = \mathbb{E}_k\left[\hat{U}_h^k(s, a) \mid \mathbb{I}_h^k(s, a) = 1\right] \cdot \mu_h^k(s, a)$$
$$+ \mathbb{E}_k\left[\hat{U}_h^k(s, a) \mid \mathbb{I}_h^k(s, a) = 0\right] \cdot (1 - \mu_h^k(s, a))$$
$$= \mathbb{E}_k\left[\frac{\sum_{h'=1}^H \ell_{h'}^k(s_{h'}^k, a_{h'}^k)}{\mu_h^k(s, a) + \gamma} \mid \mathbb{I}_h^k(s, a) = 1\right] \cdot \mu_h^k(s, a)$$
$$= \frac{\mu_h^k(s, a)}{\mu_h^k(s, a) + \gamma}\mathbb{E}_k\left[\sum_{h'=h}^H \ell_{h'}^k(s_{h'}^k, a_{h'}^k) \mid s_h^k = s, a_h^k = a\right]$$
$$= \frac{\mu_h^k(s, a)}{\mu_h^k(s, a) + \gamma} U_h^k(s, a).$$

$\square$

Given our novel regret decomposition in Equation (2) and

the lemma above, the rest of the analysis follows similar steps as those in (Luo et al., 2021).

*Proof sketch of Theorem 4.1.* Using Equation (2), we can break the regret of the algorithm as follows:

$$
\underbrace{\sum_{k,h,s} \mu_h^\star(s) \left\langle \pi_h^k(\cdot \mid s), U_h^k(s,\cdot) - \hat{U}_h^k(s,\cdot) \right\rangle}_{\text{BIAS}_1}
$$

$$
+ \underbrace{\sum_{k,h,s} \mu_h^\star(s) \left\langle \pi_h^\star(\cdot \mid s), \hat{U}_h^k(s,\cdot) - U_h^k(s,\cdot) \right\rangle}_{\text{BIAS}_2}
$$

$$
+ \underbrace{\sum_{k,h,s} \mu_h^\star(s) \left\langle \pi_h^k(\cdot \mid s) - \pi_h^\star(\cdot \mid s), \hat{U}_h^k(s,\cdot) - B_h^k(s,\cdot) \right\rangle}_{\text{REG}}
$$

$$
+ \underbrace{\sum_{k,h,s} \mu_h^\star(s) \left\langle \pi_h^k(\cdot \mid s) - \pi_h^\star(\cdot \mid s), B_h^k(s,\cdot) \right\rangle}_{\text{BONUS}}, \qquad (4)
$$

From Lemma 4.2 we immediately get that $\mathbb{E}_k[\text{BIAS}_2] \leq 0$. Since $\|\hat{U}_h^k\|_\infty \leq H/\gamma$ we can also bound with high probability $\text{BIAS}_2 \leq \tilde{O}(H^2/\gamma)$ using standard concentration bounds (see Lemma B.4). Again, using Lemma 4.2 the expectation of $\text{BIAS}_1$ equals the following,

$$
\sum_{k=1}^{K} \sum_{h,s,a} \mu_h^\star(s)\pi_h^k(a \mid s) \left( U_h^k(s,a) - \mathbb{E}_k\left[ \hat{U}_h^k(s,a) \right] \right)
$$

$$
= \sum_{k=1}^{K} \sum_{h,s,a} \mu_h^\star(s)\pi_h^k(a \mid s) U_h^k(s,a) \left( 1 - \frac{\mu_h^k(s,a)}{\mu_h^k(s,a) + \gamma} \right)
$$

$$
= \sum_{k=1}^{K} \sum_{h,s,a} \mu_h^\star(s) U_h^k(s,a) \frac{\gamma \pi_h^k(a \mid s)}{\mu_h^k(s,a) + \gamma}
$$

Bounding $U_h^k(s,a)$ by $H$ we get that $\mathbb{E}_k[\text{BIAS}_1] \leq \frac{1}{3}\sum_{k=1}^{K}\sum_{h,s} \mu_h^\star(s)b_h^k(s)$. Using a form of Freedman's inequality that takes into account the second moment of the estimator, we can also bound $\text{BIAS}_1 - \mathbb{E}_k[\text{BIAS}_1] \leq \frac{1}{3}\sum_{k=1}^{K}\sum_{h,s} \mu_h^\star(s)b_h^k(s) + \tilde{O}(H^2/\gamma)$. In total we get that,

$$
\text{BIAS}_1 \leq \frac{2}{3} \underbrace{\sum_{k=1}^{K}\sum_{h,s} \mu_h^\star(s)b_h^k(s)}_{(i)} + \tilde{O}(H^2/\gamma)
$$

Term $(i)$ is challenging due to the distribution mismatch between $\mu_h^\star(s)$ and $\mu_h^k(s)$ in the denominator of $b_h^k(s)$. However, we will later see that the BONUS term will cancel it out, essentially allowing us to replace $\mu_h^\star(s)$ with $\mu_h^k(s)$.

Before we turn to the BONUS term, let's consider REG. Using the standard entropy-regularized OMD guarantee

(Lemma E.5), REG can be bounded by,

$$
\frac{H \ln A}{\eta} + 2\eta \sum_{k,h,s,a} \mu_h^\star(s)\pi_h^k(a \mid s)\left( \hat{U}_h^k(s,a) - B_h^k(s,a) \right)^2
$$

$$
\leq \tilde{O}\left( \frac{H}{\eta} + \eta H^5 K \right) + 2\eta \sum_{k,h,s,a} \mu_h^\star(s)\pi_h^k(a \mid s)\hat{U}_h^k(s,a)^2
$$

In the inequality we've used the fact that $b_h^k(s) \leq 3H$, and thus, $B_h^k(s,a) \leq 3H^2$ since it is a $Q$-function with respect to $b_h^k(s)$. Once again, the second sum here may not be bounded due to the mismatch between $\mu_h^\star(s)$ and $\mu_h^k(s)$ in the denominator of $\hat{U}_h^k(s,a)$. However, note that $\mathbb{E}_k[\hat{U}_h^k(s,a)^2] \leq H^2/(\mu_h^k(s,a) + \gamma)$ and using standard concentration inequalities we can bound the the last term in the last display by,

$$
2\eta H^2 \sum_{k,h,s,a} \frac{\mu_h^\star(s)\pi_h^k(a \mid s)}{\mu_h^k(s,a) + \gamma} + \tilde{O}\left( \eta \frac{H^3}{\gamma^2} \right)
$$

$$
= \frac{1}{3} \underbrace{\sum_{k,h,s} \mu_h^\star(s)b_h^k(s)}_{(ii)} + \tilde{O}\left( \frac{H^2}{\gamma} \right),
$$

where we've set $\eta = \gamma/(2H)$ as in the statement of the theorem. Finally, recall that $B_h^k$ is the $Q$-function with respect to $b^k$ as losses. Applying the standard value difference lemma (Lemma 2.1) we have,

$$
\text{BONUS} = \sum_k V_1^{\pi^k}(s_{init}; b^k) - V^{\pi^\star}(s_{init}; b^k)
$$

$$
= \sum_{k,h,s} \mu_h^k(s)b_h^k(s) - \sum_{k,h,s} \mu_h^\star(s)b_h^k(s).
$$

The negative term exactly cancels out $(i)$ and $(ii)$ from $\text{BIAS}_1$ and REG. The positive term is bounded by,

$$
3\gamma H \sum_{k=1}^{K}\sum_{h,s,a} \frac{\mu_h^k(s)\pi_h^k(a \mid s)}{\mu_h^k(s)\pi_h^k(a \mid s) + \gamma} \leq 3\gamma H^2 SAK.
$$

Summing all the terms and setting $\eta$ and $\gamma$ as in the statement of the theorem completes the proof. $\qquad \square$

**Computational complexity of Algorithm 1.** The state occupancy measure can be computed using the following recursive formula:

$$
\mu_{h+1}^\pi(s') = \sum_{s,a} \mu_h^\pi(s)\pi_h(a \mid s)P(s' \mid s,a),
$$

so that the full occupancy measure can be computed in $O(HS^2A)$. The bonus $B^k$ is calculated via backward dynamic programming also in $O(HS^2A)$, and the policy computation is done in $O(HSA)$. Thus, the total complexity per iteration is $O(HS^2A)$.

# 5. Unknown Dynamics

The adaptation of our algorithm to the unknown dynamics case is relatively straightforward. Since we do not know the transition function, we cannot compute $\mu^k$ and can't perform the Bellman backup in the computation of $B^k$. As standard in the literature, we employ Bernstein-style confidence sets for the transition function (e.g. Shani et al. (2020); Luo et al. (2021); Jin et al. (2020)). Specifically, let $\mathcal{P}^k = \{\mathcal{P}_h^k(s,a)\}_{s,a,h}$ such that $p_h'(\cdot \mid s,a) \in \mathcal{P}_h^k(s,a)$ if and only if $p_h'(\cdot \mid s,a) \in \Delta_\mathcal{S}$ and for every $s' \in \mathcal{S}$:

$$
\begin{aligned}
&|p_h'(s' \mid s,a) - \bar{p}_h^k(s' \mid s,a)| \\
&\le 4\sqrt{\frac{\bar{p}_h^k(s' \mid s,a)\log\frac{10HSAK}{\delta}}{n_h^k(s,a)\vee 1}} + \frac{10\log\frac{10HSAK}{\delta}}{n_h^k(s,a)\vee 1},
\end{aligned}
$$

where $n_h^k(s,a,s')$ is the number of times we visited $s,a$ at time $h$ and transitioned to $s'$, $n_h^k(s,a) = \sum_{s'} n_h^k(s,a,s')$ and $\bar{p}_h^k(s' \mid s,a) = n_h^k(s,a,s')/n_h^k(s,a)$ is the empirical transition probability. Based on $\mathcal{P}^k$, we replace $\mu_h^k(s)$ in the definition of $\hat{U}_h^k(s,a)$ with an upper confidence bound $\overline{\mu}_h^k(s) := \max_{p' \in \mathcal{P}^k} \mu_h^{\pi^k,p'}(s)$ (see Algorithm 3). Here, the notation $\mu_h^{\pi^k,p'}(s)$ represents the occupancy measure with respect to the transition $p'$ (instead of $p$). The local bonus $b_h^k(s)$ is adapted correspondingly and is exactly as in Luo et al. (2021) - see definition in Algorithm 2. Finally, the bonus $B^k$ is estimated optimistically based on the transition $p' \in \mathcal{P}^k$ that maximizes it, as described in the update computation of $\hat{B}^k$ in Algorithm 2.

The regret guarantee of our algorithm is established in the following theorem.

**Theorem 5.1.** *Under unknown dynamics, running Algorithm 2 with $\eta = (H\sqrt{SAK} + H^2\sqrt{K})^{-1}$ and $\gamma = 2\eta H$, guarantees with probability $1 - \delta$,*

$$
R_K \le \tilde{O}(H^3 S\sqrt{AK} + H^4 S^3 A).
$$

The second term of the regret is typically of low order for sufficiently large $K$. The above regret improves upon Cohen et al. (2021b) by a factor of $H^2 S^5 A^2$ and provides a high-probability regret bound instead of an expected one. Cohen et al. (2021b) uses a reduction to distorted linear bandits and requires solving a convex optimization problem in each iteration. In contrast, our algorithm benefits from a more computationally efficient, closed-form update. Our bound also improves upon Efroni et al. (2021) and Cassel et al. (2024) for the case of tabular MDPs by a factor of $SA\sqrt{H}$ even though they consider stochastic i.i.d losses as opposed to adversarial losses in our setting. The gap from the lower bound is a factor $H\sqrt{S}$, but it is worth noting that even in the semi-bandit case, the best-known regret is $H^2 S\sqrt{AK}$ (Jin et al., 2020) which is achieved by a less efficient occupancy-measure-based algorithm. In fact, even though we consider

---

**Algorithm 2** Policy Optimization with Aggregated Bandit Feedback and Unknown Transition Function

**Input:** state space $\mathcal{S}$, action space $\mathcal{A}$, horizon $H$, learning rate $\eta > 0$, exploration parameter $\gamma > 0$, confidence parameter $\delta > 0$
**Initialization:** Set $\pi_h^1(a \mid s) = 1/A$ for every $(h,s,a)$
**for** $k = 1, 2, \ldots, K$ **do**

  Play episode $k$ with policy $\pi^k$ and observe aggregated bandit feedback $L_{1:H}^k = \sum_{h=1}^H \ell_h^k(s_h^k, a_h^k)$
  $\overline{\mu}_h^k(s) = \max_{p' \in \mathcal{P}^k} \mu_h^{\pi^k,p'}(s)$
  $\underline{\mu}_h^k(s) = \min_{p' \in \mathcal{P}^k} \mu_h^{\pi^k,p'}(s)$
  $\hat{U}_h^k(s,a) = \frac{L_{1:H}^k}{\overline{\mu}_h^k(s,a)+\gamma}\mathbb{I}_h^k(s,a)$
  # Bonus Computation
  Set $\hat{B}_{H+1}^k(s,a) = 0$ for every $(s,a) \in \mathcal{S}\times\mathcal{A}$.
  **for** $h = H, H-1, \ldots, 1$ **do**
    **for** $(s,a) \in \mathcal{S}\times\mathcal{A}$ **do**
      $\tilde{b}_h^k(s) = \sum_{a\in\mathcal{A}} \frac{3\gamma H\pi_h^k(a|s)}{\overline{\mu}_h^k(s)\pi_h^k(a|s)+\gamma}$
      $\bar{b}_h^k(s) = \sum_{a\in\mathcal{A}} \frac{H\pi_h^k(a|s)(\overline{\mu}_h^k(s,a)-\underline{\mu}_h^k(s,a))}{\overline{\mu}_h^k(s)\pi_h^k(a|s)+\gamma}$
      $b_h^k(s) = \tilde{b}_h^k(s) + \bar{b}_h^k(s)$

      $\hat{B}_h^k(s,a) = b_h^k(s)$
      $\quad + \max_{p'\in\mathcal{P}_h^k(s,a)} \sum_{s'\in\mathcal{S},a'\in\mathcal{A}} p_h'(s' \mid s,a)\pi_{h+1}^k(a' \mid s')$
      $\qquad\qquad\qquad\qquad\qquad\quad \cdot \hat{B}_{h+1}^k(s',a')$

    **end for**
  **end for**
  # Policy Improvement
  For every $(s,a,h) \in \mathcal{S}\times\mathcal{A}\times[H]$ by:

  $$
  \pi_h^{k+1}(a \mid s) = \frac{\pi_h^k(a \mid s)e^{-\eta(\hat{U}_h^k(s,a)-\hat{B}_h^k(s,a))}}{\sum_{a'}\pi_h^k(a' \mid s)e^{-\eta(\hat{U}_h^k(s,a')-\hat{B}_h^k(s,a'))}}.
  $$

**end for**

---

less informative feedback, we match the state-of-the-art regret for PO with semi-bandit feedback (Luo et al., 2021).

The proof of Theorem 5.1 follows similar steps as in the known dynamics case, but includes additional steps to control the transition estimation error. These utilize standard techniques that we highlight in the following proof sketch.

*Proof sketch.* We utilize the same regret decomposition as in Equation (4) but replace $B^k$ with $\hat{B}^k$ in the Reg and Bonus terms. The estimator $\hat{U}^k$ remains optimistic (i.e., in expectation it is smaller than $U^k$) since with high probability $\overline{\mu}_h^k(s) \ge \mu_h^k(s)$ for any $h,s$, and $k$. Thus, $\text{BIAS}_2 \le \tilde{O}(H^2/\gamma)$ with high probability in a similar way to the known dynamics case. In $\text{Bias}_1$, on the other hand,

using $\overline{\mu}_h^k(s)$ instead of $\mu_h^k(s)$ introduces additional bias of order of,

$$\sum_{k,h,s,a} \mu_h^\star(s)\pi_h^k(a \mid s)H\frac{\overline{\mu}_h^k(s,a) - \mu_h^k(s,a)}{\overline{\mu}_h^k(s,a) + \gamma}$$

$$\leq \sum_{k,h,s,a} \mu_h^\star(s)\pi_h^k(a \mid s)H\frac{\overline{\mu}_h^k(s,a) - \underline{\mu}_h^k(s,a)}{\overline{\mu}_h^k(s,a) + \gamma},$$

where the inequality follows from the fact that $\underline{\mu}_h^k(s,a) \leq \mu_h^k(s,a)$ w.h.p. Note that this is exactly $\sum_{k,h,s,a} \mu_h^\star(s)\overline{b}_h^k(s)$. This term is in addition to the terms we already had in $\texttt{Bias}_1$ in the known dynamics case. In total we have,

$$\texttt{BIAS}_1 \leq \frac{2}{3} \sum_{k,h,s} \mu_h^\star(s)\tilde{b}_h^k(s) + \sum_{k,h,s} \mu_h^\star(s)\overline{b}_h^k(s) + \tilde{O}\left(\frac{H^2}{\gamma}\right)$$

$\texttt{Reg}$ is bounded in a similar way to the known dynamics case. Given that the estimator is optimistic and exhibits lower variance when using the upper confidence bound on the occupancy measure, we get that $\texttt{Reg}$ is bounded by,

$$\frac{H \ln A}{\eta} + 16\eta H^5 K + \frac{1}{3} \sum_{k,h,s} \mu_h^\star(s)\tilde{b}_h^k(s) + \tilde{O}\left(\frac{H^2}{\gamma}\right).$$

Finally, note $\hat{B}^k$ is not an exact $Q$-function since we don't know the actual transition function. Thus, we can not directly use the value difference lemma to show that $\texttt{Bonus} = \sum_{k,h,s} \mu_h^k(s)b_h^k(s) - \sum_{k,h,s} \mu_h^\star(s)b_h^k(s)$. However, using the fact that we update $\hat{B}$ with a transition function in the confidence set that maximizes it, we are able to show that w.h.p,

$$\texttt{BONUS} \leq \sum_{k,h,s} \overline{\mu}_h^k(s)b_h^k(s) - \sum_{k,h,s} \mu_h^\star(s)b_h^k(s).$$

Once again, the negative term above cancels the terms that depend on $b^k$ in $\texttt{Bias}_1$ and $\texttt{Reg}$. Recall that $b_h^k(s) = \tilde{b}_h^k(s) + \overline{b}_h^k(s)$. Exactly as in the known dynamics case, $\sum_{k,h,s} \overline{\mu}_h^k(s)\tilde{b}_h^k(s) \leq O(\gamma H^2 SAK)$. The term $\sum_{k,h,s} \overline{\mu}_h^k(s)\overline{b}_h^k(s)$ equals to,

$$H \sum_{k,h,s} \overline{\mu}_h^k(s,a)\frac{\overline{\mu}_h^k(s,a) - \underline{\mu}_h^k(s,a)}{\overline{\mu}_h^k(s,a) + \gamma}$$

$$\leq H \sum_{k,h,s} |\overline{\mu}_h^k(s,a) - \underline{\mu}_h^k(s,a)|.$$

The last is a standard transition estimation error and is bounded by $\tilde{O}(H^3 S\sqrt{AK} + H^3 S^3 A)$ (Jin et al., 2020). Summing all the terms and setting $\eta$ and $\gamma$ as in the statement of the theorem completes the proof. □

**Computational complexity of Algorithm 2.** The upper/lower confidence occupancy measure computation can be done using (Jin et al., 2020, Algorithm 3), with a complexity of $O(HS^2A)$ per state. The same can be done for the computation of the bonus $\hat{B}^k$. The policy update is done in $O(HSA)$. Thus, the total complexity per iteration is $O(HS^3A)$.

## 6. Lower Bound

Our lower bound uses a lower bound for the multi-task bandit problem (see for example Lattimore & Szepesvári (2020)). In the multitask bandit problem, the learner faces simultaneously $H$ instances of the $A$-armed bandit problem. At each round $k \in [K]$, the learner selects $H$ actions, one for each bandit problem, and observes the sum of the losses associated with these $H$ actions. This scenario can be seen as analogous to MDPs with a single state (i.e., $S = 1$) and aggregate bandit feedback. This is due to the fact that whenever we have only a single state no information is gained within the episode. (Recall that the losses are horizon-dependent, so we have $\ell_h(s_0, a)$, for each action $a$, time $h \in [H]$ and using the single state $s_0$.)

**Lemma 6.1** (Theorem 1 in Cohen et al. (2017)). *Assume that $A \geq 2$. Any learning algorithm for the multi-task bandit problem must incur at least $\tilde{\Omega}(H^2\sqrt{AK})$ expected regret in the worst case.*

With that in hand, we obtain the following for online MDPs with aggregate bandit feedback.

**Theorem 6.2.** *Assume that $H, S, A \geq 2$ and $K \geq 2S$. Any learning algorithm for the online MDPs with known dynamics and aggregate bandit feedback problem must incur at least $\Omega(H^2\sqrt{SAK})$ expected regret in the worst case.*

The above lower bound shows that our regret upper bound for the known dynamics case is tight up to poly-logarithmic factors whenever $\sqrt{SA} \geq H$. For unknown dynamics, clearly the same lower bound holds, and we have a multiplicative gap of $H\sqrt{S}$. We note that determining the exact optimal bound for the unknown dynamics case remains an open problem, both with aggregate bandit feedback as well as with the well-studied semi-bandit feedback.

Due to space limitations, the proof is deferred to Appendix D. At a high level, the proof is constructed as follows: Consider an MDP where in the first step of each episode, the agent transitions to each state with an equal probability of $1/S$, regardless of the chosen action. From the second step onward, the agent remains in the same state for the remainder of the episode, wherein a hard multi-task bandit problem is encoded in each state. Roughly speaking, each state is visited approximately $K/S$ times. Using Lemma 6.1, the regret from visits to each state is approximately $H^2\sqrt{AK/S}$. Summing the regret across all states gives a lower bound of

$H^2\sqrt{SAK}$.

## 7. Discussion and Future Work

In this paper, we introduced the concept of $U$-functions, which allows us to establish the first regret bounds for online MDPs with aggregate bandit feedback using PO algorithms. One of the advantages of the PO framework is its natural extension to function approximation (Luo et al., 2021; Sherman et al., 2023; Dai et al., 2023; Liu et al., 2024).

It would be interesting to see whether the $U$-function concept could be useful in achieving regret bounds with aggregate bandit feedback for environments with infinitely many states under a function approximation assumption. We note that the main challenge in extending our results to linear MDPs, for example, is that the $U$-function is not linear under this assumption. Still, it is possible that in such settings, the $U$-function would have certain properties that would allow achieving sub-linear regret.

Moreover, it would be interesting to study the incorporation of the $U$-function in deep reinforcement learning algorithms and explore whether it can provide practical benefits or improved performance in large-scale, real-world RL problems with aggregate bandit feedback.

## Acknowledgments

This project has received funding from the European Research Council (ERC) under the European Union's Horizon 2020 research and innovation program (grant agreement No. 882396), by the Israel Science Foundation, the Yandex Initiative for Machine Learning at Tel Aviv University and a grant from the Tel Aviv University Center for AI and Data Science (TAD).

## Impact Statement

This paper presents work whose goal is to advance the field of Machine Learning. There are many potential societal consequences of our work, none which we feel must be specifically highlighted here.

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

## A. Summery of notations

For convenience, the table below summarizes most of the notation used throughout the paper.

| | |
|---|---|
| $H$ | Horizon length of each episode |
| $K$ | Total number of episodes |
| $S$ | Number of states in the MDP |
| $A$ | Number of actions in the MDP |
| $p$ | Transition function of the MDP |
| $\ell^k$ | Loss function in episode $k$ |
| $R_K$ | Cumulative regret over $K$ episodes |
| $V_h^\pi(s;\ell)$ | Value function at state $s$ and time $h$ under policy $\pi$ and loss function $\ell$ |
| $V_h^k(s)$ | Value under policy $\pi^k$ and loss function $\ell^k$. I.e., $V_h^k(s) = V_h^{\pi^k}(s;\ell^k)$ |
| $Q_h^\pi(s,a;\ell)$ | $Q$-function at state $s$, action $a$, and time $h$ under policy $\pi$ and loss function $\ell$ |
| $Q_h^k(s,a)$ | $Q$-function under policy $\pi^k$ and loss function $\ell^k$. I.e., $Q_h^k(s,a) = Q_h^{\pi^k}(s,a;\ell^k)$ |
| $U_h^\pi(s,a;\ell)$ | Expected total loss of the episode, conditioned on taking action $a$ in state $s$ at time $h$ under policy $\pi$ |
| $U_h^k(s,a)$ | Conditional expected total loss with respect to $\pi^k$ and $\ell^k$. I.e., $U_h^k(s,a) = U_h^{\pi^k}(s,a;\ell^k)$ |
| $W_h^\pi(s;\ell)$ | Expected cumulative loss up to time $h-1$, conditioned on reaching state $s$ at time $h$ under policy $\pi$ |
| $W_h^k(s)$ | Conditional expected loss with respect to $\pi^k$ and $\ell^k$. I.e., $W_h^k(s) = W_h^{\pi^k}(s;\ell^k)$ |
| $\mu_h^\pi(s,a)$ | Occupancy measure: probability of being in state $s$ and taking action $a$ at time $h$ under policy $\pi$ |
| $\mu_h^k(s,a)$ | Occupancy measure under $\pi^k$: $\mu_h^k(s,a) = \mu_h^{\pi^k}(s,a)$ |
| $\mu_h^{\pi,p'}(s)$ | Occupancy measure with respect to a transition function $p'$ |
| $\mathbb{E}_k[\cdot]$ | Expectation condition on the policy $\pi^k$: $\mathbb{E}_k[\cdot] = \mathbb{E}[\cdot \mid \pi^k]$ |
| $\mathbb{I}_h^k(s,a)$ | The indicator for visiting $s$ and taking $a$ at time $h$ in episode $k$: $\mathbb{I}_h^k(s,a) = \mathbb{I}\{s_h^k = s, a_h^k = a\}$ |
| $\hat{U}_h^k(s,a)$ | Estimator of $U_h^k(s,a)$ - See Algorithms 1 and 2 |
| $b_h^k(s)$ | Intermediate bonus function - See Algorithms 1 and 2 |
| $B_h^k(s,a)$ | Bonus function, defined as the $Q$-function with respect to $b^k$ - See Algorithm 1 |
| $\hat{B}_h^k(s,a)$ | Estimator of $B_h^k(s,a)$ - See Algorithm 2 |
| $\mathcal{P}^k$ | Confidence set of transition functions at episode $k$ |
| $\overline{\mu}_h^k(s,a)$ | Upper confidence bound on the occupancy measure - See Algorithm 2 |
| $\underline{\mu}_h^k(s,a)$ | Lower confidence bound on the occupancy measure - See Algorithm 2 |
| $\iota$ | A logarithmic factor of $\log \frac{HSAK}{\delta}$ |

## B. Known Dynamics

**Theorem B.1** (Restatement of Theorem 4.1)**.** *Under known dynamics, running Algorithm 1 with* $\eta = \frac{\sqrt{\iota}}{H\sqrt{SAK}+H^2\sqrt{K}}$ *and* $\gamma = 2\eta H$ *for* $\iota = \log\frac{HSAK}{\delta}$, *guarantees with probability* $1 - \delta$,

$$R_K \lesssim H^2\sqrt{SAK\iota} + H^3\sqrt{K\iota}.$$

*Proof.* Using Equation (2), we can break the regret of the algorithm as,

$$
\begin{aligned}
R_K &= \sum_{k=1}^{K}\sum_{h,s}\mu_h^\star(s)\left\langle \pi_h^k(\cdot\mid s) - \pi_h^\star(\cdot\mid s), U_h^k(s,\cdot)\right\rangle \\
&= \underbrace{\sum_{k=1}^{K}\sum_{h,s}\mu_h^\star(s)\left\langle \pi_h^k(\cdot\mid s), U_h^k(s,\cdot) - \hat{U}_h^k(s,\cdot)\right\rangle}_{\text{BIAS}_1} + \underbrace{\sum_{k=1}^{K}\sum_{h,s}\mu_h^\star(s)\left\langle \pi_h^\star(\cdot\mid s), \hat{U}_h^k(s,\cdot) - U_h^k(s,\cdot)\right\rangle}_{\text{BIAS}_2} \\
&+ \underbrace{\sum_{k=1}^{K}\sum_{h,s}\mu_h^\star(s)\left\langle \pi_h^k(\cdot\mid s) - \pi_h^\star(\cdot\mid s), \hat{U}_h^k(s,\cdot) - B_h^k(s,\cdot)\right\rangle}_{\text{REG}} + \underbrace{\sum_{k=1}^{K}\sum_{h,s}\mu_h^\star(s)\left\langle \pi_h^k(\cdot\mid s) - \pi_h^\star(\cdot\mid s), B_h^k(s,\cdot)\right\rangle}_{\text{BONUS}},
\end{aligned}
$$

Due to the optimism of the estimator $\text{BIAS}_2 \le \tilde{O}(H^2/\gamma)$ under the good event $G_3$ which holds with high probability (see Lemma B.4). In Lemmas B.6 to B.8 we bound $\text{BIAS}_1$, REG and BONUS. Overall we get that,

$$R_K \lesssim \frac{H\ln A}{\eta} + \eta H^5 K + \gamma H^2 SAK + \frac{H^2\iota}{\gamma}$$

Plugging $\eta$ and $\gamma$ completes the proof.

$\square$

### B.1. Good event

We define the following good event $G = \bigcap_{i=1}^{3} G_i$ which holds with high probability (Lemma B.5):

$$
G_1 = \left\{ \sum_{k=1}^{K}\sum_{h,s,a}\mu_h^\star(s)\pi_h^k(a\mid s)\left(\mathbb{E}_k\left[\hat{U}_h^k(s,a)\right] - \hat{U}_h^k(s,a)\right) \le \frac{1}{3}\sum_{k=1}^{K}\sum_{h,s}\mu_h^\star(s)b_h^k(s) + \frac{H^2\log\frac{6}{\delta}}{\gamma} \right\}
$$

$$
G_2 = \left\{ \sum_{k=1}^{K}\sum_{h,s,a}\frac{\mu_h^\star(s)\pi_h^k(a\mid s)}{\mu_h^k(s,a)+\gamma}\left(\frac{\mathbb{I}\{s_h^k = s, a_h^k = a\}}{\mu_h^k(s,a)+\gamma} - 1\right) \le \frac{H\ln\frac{6H}{\delta}}{2\gamma^2} \right\}
$$

$$
G_3 = \left\{ \sum_{k=1}^{K}\sum_{h,s}\mu_h^\star(s)\left\langle \pi_h^\star(\cdot\mid s), \hat{U}_h^k(s,\cdot) - U_h^k(s,\cdot)\right\rangle \le \frac{H^2}{2\gamma}\ln\frac{6H}{\delta} \right\}
$$

Under the good event the regret will be deterministically bounded.

**Lemma B.2** (Event $G_1$)**.** *With probability* $1 - \delta$,

$$\sum_{k=1}^{K}\sum_{h,s,a}\mu_h^\star(s)\pi_h^k(a\mid s)\left(\mathbb{E}_k\left[\hat{U}_h^k(s,a)\right] - \hat{U}_h^k(s,a)\right) \le \frac{1}{3}\sum_{k=1}^{K}\sum_{h,s}\mu_h^\star(s)b_h^k(s) + \frac{H^2\log\frac{1}{\delta}}{\gamma}$$

*Proof.* Let $Y_k = \sum_{h,s}\mu_h^\star(s)\left\langle \pi_h^k(\cdot\mid s), \hat{U}_h^k(s,\cdot)\right\rangle$. Note that $\mathbb{E}_k[Y_k] - Y_k$ is a martingale difference sequence. To bound $\sum_k \mathbb{E}_k[Y_k] - \sum_k Y_k$ we'll use a form of Freedman's Inequality (Lemma E.2). For that, we need to bound the absolute value

of the difference and its variance. For the absolute value of the difference we have $|\mathbb{E}_k[Y_k] - Y_k| \le \max\{\mathbb{E}_k[Y_k], Y_k\} \le \frac{H^2}{\gamma}$. For the variance, note that it is bounded by the second moment of $Y_t$: $\mathbb{E}_k[(\mathbb{E}_k[Y_k] - Y_k)^2] \le \mathbb{E}_k[Y_k^2]$. We further bound $\mathbb{E}_k[Y_k^2]$ as follows:

$$
\begin{aligned}
\mathbb{E}_k[Y_k^2] = \mathbb{E}_k &\left[ \left( \sum_{h,s,a} \mu_h^\star(s)\pi_h^k(a \mid s)\hat{U}_h^k(s,a) \right)^2 \right] \\
\le \mathbb{E}_k &\left[ \left( \sum_{h,s,a} \mu_h^\star(s)\pi_h^k(a \mid s) \right) \left( \sum_{h,s,a} \mu_h^\star(s)\pi_h^k(a \mid s) \left( \hat{U}_h^k(s,a) \right)^2 \right) \right] && \text{(Cauchy–Schwarz inequality)} \\
= H\mathbb{E}_k &\left[ \sum_{h,s,a} \mu_h^\star(s)\pi_h^k(a \mid s) \left( \hat{U}_h^k(s,a) \right)^2 \right] \\
= H\mathbb{E}_k &\left[ \sum_{h,s,a} \mu_h^\star(s)\pi_h^k(a \mid s) \frac{(L_{1:H}^k)^2}{(\mu_h^k(s,a)+\gamma)^2} \mathbb{I}_h^k(s,a) \right] \\
\le H^3\mathbb{E}_k &\left[ \sum_{h,s,a} \mu_h^\star(s)\pi_h^k(a \mid s) \frac{\mathbb{I}_h^k(s,a)}{(\mu_h^k(s,a)+\gamma)^2} \right] && (L_{1:H}^k \le H) \\
= H^3 &\sum_{h,s,a} \mu_h^\star(s)\pi_h^k(a \mid s) \frac{\mu_h^k(s,a)}{(\mu_h^k(s,a)+\gamma)^2} && (\mathbb{E}_k[\mathbb{I}_h^k(s,a)] = \mu_h^k(s,a)) \\
\le H^3 &\sum_{h,s,a} \mu_h^\star(s) \frac{\pi_h^k(a \mid s)}{\mu_h^k(s,a)+\gamma}.
\end{aligned}
$$

Using Lemma E.2 with probability $1 - \delta$,

$$
\sum_k \mathbb{E}_k[Y_k] - \sum_k Y_k \le \alpha \sum_k \mathbb{E}_k[Y_k^2] + \frac{\log \frac{1}{\delta}}{\alpha}
$$

where $\alpha \in (0, 1/R]$ for $R$ such that $|\mathbb{E}_k[Y_k] - Y_k| \le R$ for any $k$. In our case, $|\mathbb{E}_k[Y_k] - Y_k| \le \frac{H^2}{\gamma}$. And so,

$$
\sum_k \mathbb{E}_k[Y_k] - \sum_k Y_k \le \frac{1}{3} \sum_{k=1}^K \sum_{h,s} \mu_h^\star(s)b_h^k(s) + \frac{H^2 \log \frac{1}{\delta}}{\gamma}.
$$

$\square$

**Lemma B.3** (Event $G_2$). *With probability $1 - \delta$,*

$$
\sum_{k,h,s,a} \frac{\mu_h^\star(s)\pi_h^k(a \mid s)}{\mu_h^k(s,a)+\gamma} \left( \frac{\mathbb{I}\{s_h^k = s, a_h^k = a\}}{\mu_h^k(s,a)+\gamma} - 1 \right) \le \frac{H \ln \frac{H}{\delta}}{2\gamma^2}
$$

*Proof.* Follows directly from Lemma E.4 with $Z_h^k(s,a) = z_h^k(s,a) = \frac{\mu_h^\star(s)\pi_h^k(a|s)}{\mu_h^k(s,a)+\gamma} \le \frac{1}{\gamma}$ and $\tilde{\mu}_h^k(s,a) = \mu_h^k(s,a)$. $\square$

**Lemma B.4** (Event $G_3$). *With probability $1 - \delta$,*

$$
\sum_{k=1}^K \sum_{h,s} \mu_h^\star(s) \left\langle \pi_h^\star(\cdot \mid s), \hat{U}_h^k(s,\cdot) - U_h^k(s,\cdot) \right\rangle \le \frac{H^2}{2\gamma} \ln \frac{H}{\delta}
$$

*Proof.* By invoking Lemma E.4 with $Z_h^k(s,a) = \mu_h^\star(s)\pi_h^k(a,s)[\mathbb{I}_h^k(s,a)L_{1:H}^k + (1 - \mathbb{I}_h^k(s,a))U_h^k(s,a)] \leq H$, $z_h^k(s,a) = \mathbb{E}_k[Z_h^k(s,a)] = \mu_h^k(s)\pi_h^k(a,s)U_h^k(s,a)$ and $\tilde{\mu}_h^k(s,a) = \mu_h^k(s,a)$, we get,

$$
\sum_{k=1}^K \sum_{h,s} \mu_h^\star(s) \left\langle \pi_h^\star(\cdot \mid s), \hat{U}_h^k(s,\cdot) - U_h^k(s,\cdot) \right\rangle
$$

$$
= \sum_{k=1}^K \sum_{h,s,a} \frac{\mu_h^\star(s)\pi_h^k(a,s)\mathbb{I}_h^k(s,a)L_{1:H}^k}{\mu_h^k(s,a) + \gamma} - \sum_{k=1}^K \sum_{h,s,a} \mu_h^\star(s)\pi_h^k(a,s)U_h^k(s,a) \leq \frac{H^2}{2\gamma} \ln \frac{H}{\delta}
$$

$\square$

**Lemma B.5.** *Under Algorithm 1, the good event $G = \bigcap_{i=1}^3 G_i$ holds with probability of at least $1 - \delta$.*

*Proof.* The proof directly follows from invoking Lemmas B.2 to B.4 with $\delta/3$ and taking the union bound. $\square$

## B.2. Bound on BIAS$_1$

**Lemma B.6.** *Under the good event $G_1$,*

$$
\text{BIAS}_1 \leq \frac{2}{3} \sum_{k=1}^K \sum_{h,s} \mu_h^\star(s)b_h^k(s) + \frac{H^2 \log \frac{6}{\delta}}{\gamma}.
$$

Let $Y_k = \sum_{h,s} \mu_h^\star(s) \left\langle \pi_h^k(\cdot \mid s), \hat{U}_h^k(s,\cdot) \right\rangle$

$$
\text{BIAS}_1 = \sum_{k=1}^K \sum_{h,s} \mu_h^\star(s) \left\langle \pi_h^k(\cdot \mid s), U_h^k(s,\cdot) \right\rangle - \sum_k \mathbb{E}_k[Y_k] + \sum_k \mathbb{E}_k[Y_k] - \sum_k Y_k
$$

Under the good event $G_1$, $\sum_k \mathbb{E}_k[Y_k] - \sum_k Y_k \leq \frac{1}{3}\sum_{k=1}^K \sum_{h,s} \mu_h^\star(s)b_h^k(s) + \frac{H^2 \log \frac{6}{\delta}}{\gamma}$. Using Lemma 4.2,

$$
\begin{aligned}
\sum_{h,s} \mu_h^\star(s) \left\langle \pi_h^k(\cdot \mid s), U_h^k(s,\cdot) \right\rangle - \mathbb{E}_k[Y_k] &= \sum_{k=1}^K \sum_{h,s,a} \mu_h^\star(s)\pi_h^k(a \mid s) \left( U_h^k(s,a) - \mathbb{E}_k\left[\hat{U}_h^k(s,a)\right] \right) \\
&= \sum_{k=1}^K \sum_{h,s,a} \mu_h^\star(s)\pi_h^k(a \mid s)U_h^k(s,a) \left( 1 - \frac{\mu_h^k(s,a)}{\mu_h^k(s,a) + \gamma} \right) \\
&\leq H \sum_{k=1}^K \sum_{h,s,a} \mu_h^\star(s)\pi_h^k(a \mid s) \left( 1 - \frac{\mu_h^k(s,a)}{\mu_h^k(s,a) + \gamma} \right) \qquad (U_h^k(s,a) \leq H) \\
&= \sum_{k=1}^K \sum_{h,s,a} \mu_h^\star(s)\frac{\gamma H \pi_h^k(a \mid s)}{\mu_h^k(s,a) + \gamma} \\
&= \frac{1}{3} \sum_{k=1}^K \sum_{h,s} \mu_h^\star(s)b_h^k(s).
\end{aligned}
$$

## B.3. Bound on REG

**Lemma B.7.** *For $\eta \leq \frac{\gamma}{2H}$, under the good $G_2$,*

$$
\text{REG} \leq \frac{H \ln A}{\eta} + 9\eta H^5 K + \frac{1}{3} \sum_{k,h,s} \mu_h^\star(s)b_h^k(s) + O\left( \frac{H^2\iota}{\gamma} \right)
$$

*Proof.* Using standard entropy regularized OMD guarantee (Lemma E.5),

$$\text{REG} \leq \frac{H \ln A}{\eta} + 2\eta \sum_{k=1}^{K} \sum_{h,s,a} \mu_h^\star(s) \pi_h^k(a \mid s) \left( \hat{U}_h^k(s,a) - B_h^k(s,a) \right)^2$$

$$\leq \frac{H \ln A}{\eta} + 2\eta \sum_{k=1}^{K} \sum_{h,s,a} \mu_h^\star(s) \pi_h^k(a \mid s) \hat{U}_h^k(s,a)^2 + 9\eta H^5 K,$$

where the second inequality is since $b_h^k(s) \leq 3H$ and thus $B_h^k(s,a) \leq 3H^2$. For the middle term,

$$2\eta \sum_{k,h,s,a} \mu_h^\star(s) \pi_h^k(a \mid s) \hat{U}_h^k(s,a)^2 \leq 2\eta \sum_{k,h,s,a} \mu_h^\star(s) \pi_h^k(a \mid s) \frac{H^2 \mathbb{I}\{s_h^k = s, a_h^k = a\}}{(\mu_h^k(s,a) + \gamma)^2} \qquad (L_{1:H}^k \leq H)$$

$$\leq 2\eta H^2 \sum_{k,h,s,a} \frac{\mu_h^\star(s) \pi_h^k(a \mid s)}{\mu_h^k(s,a) + \gamma} + O\left( \eta \frac{H^3 \iota}{\gamma^2} \right) \qquad \text{(Under event } G_2\text{)}$$

$$= \frac{2\eta}{3\gamma} H \sum_{k,h,s} \mu_h^\star(s) b_h^k(s) + O\left( \eta \frac{H^3 \iota}{\gamma^2} \right)$$

$$\leq \frac{1}{3} \sum_{k,h,s} \mu_h^\star(s) b_h^k(s) + O\left( \frac{H^2 \iota}{\gamma} \right).$$

The last inequality is since $\eta \leq \frac{H}{2\gamma}$ as in the statement of the lemma. $\qquad \square$

## B.4. Bound on BONUS

**Lemma B.8.** *It holds that,*

$$\text{BONUS} \leq 3\gamma H^2 SAK - \sum_{k,h,s} \mu_h^\star(s) b_h^k(s).$$

*Proof.* Recall that $B_h^k$ is the $Q$-function of policy $\pi^k$ with respect to the cost function $b^k$. Hence, by the value difference difference lemma (Lemma 2.1),

$$\text{BONUS} = \sum_{k=1}^{K} \sum_{h,s} \mu_h^\star(s) \left\langle \pi_h^k(\cdot \mid s) - \pi_h^\star(\cdot \mid s), B_h^k(s,\cdot) \right\rangle = \sum_k V_1^{\pi^k}(s_{init}; b^k) - V^{\pi^\star}(s_{init}; b^k)$$

$$= \sum_{k,h,s} \mu_h^k(s) b_h^k(s) - \sum_{k,h,s} \mu_h^\star(s) b_h^k(s).$$

For last,

$$\sum_{k=1}^{K} \sum_{h,s} \mu_h^k(s) b_h^k(s) = 3\gamma H \sum_{k=1}^{K} \sum_{h,s,a} \frac{\mu_h^k(s) \pi_h^k(a \mid s)}{\mu_h^k(s) \pi_h^k(a \mid s) + \gamma} \leq 3\gamma H^2 SAK.$$

$\qquad \square$

# C. Unknown Dynamics

---

**Algorithm 3** Policy Optimization with Aggregated Bandit Feedback and Unknown Transition Function
(detailed version of Algorithm 2)

---

**Input:** state space $\mathcal{S}$, action space $\mathcal{A}$, horizon $H$, learning rate $\eta > 0$, exploration parameter $\gamma > 0$, confidence parameter $\delta > 0$.

**Initialization:** Set $\pi_h^1(a \mid s) = \frac{1}{A}$ and visit counters $n_h^1(s, a, s') = 0$, $n_h^1(s, a) = 0$ for every $(h, s, a, s') \in [H] \times \mathcal{S} \times \mathcal{A} \times \mathcal{S}$.

**for** $k = 1, 2, \dots, K$ **do**

    Play episode $k$ with policy $\pi^k$ and observe aggregated bandit feedback $L^k = \sum_{h=1}^H \ell_h^k(s_h^k, a_h^k)$.

    Update visit counters for every $(h, s, a, s') \in [H] \times \mathcal{S} \times \mathcal{A} \times \mathcal{S}$:

$$n_h^k(s, a, s') = n_h^{k-1}(s, a, s') + \mathbb{I}_h^{k-1}(s, a, s'); \qquad n_h^k(s, a) = n_h^{k-1}(s, a) + \mathbb{I}_h^{k-1}(s, a).$$

    Compute empirical transition function $\bar{p}_h^k(s' \mid s, a) = \frac{n_h^k(s,a,s')}{\max\{n_h^k(s,a),1\}}$ and confidence set $\mathcal{P}^k = \{\mathcal{P}_h^k(s, a)\}_{s,a,h}$ such that $p_h'(\cdot \mid s, a) \in \mathcal{P}_h^k(s, a)$ if and only if $\sum_{s'} p_h'(s' \mid s, a) = 1$ and for every $s' \in \mathcal{S}$:

$$|p_h'(s' \mid s, a) - \bar{p}_h^k(s' \mid s, a)| \le 4\sqrt{\frac{\bar{p}_h^k(s' \mid s, a) \log \frac{10HSAK}{\delta}}{n_h^k(s, a) \vee 1}} + \frac{10 \log \frac{10HSAK}{\delta}}{n_h^k(s, a) \vee 1}.$$

    Compute occupancy measures $\overline{\mu}_h^k(s) = \max_{p' \in \mathcal{P}^k} \mu_h^{\pi^k, p'}(s)$ and $\underline{\mu}_h^k(s) = \min_{p' \in \mathcal{P}^k} \mu_h^{\pi^k, p'}(s)$.

    # Policy Evaluation

    $\hat{U}_h^k(s, a) = \frac{L_{1:H}^k}{\overline{\mu}_h^k(s,a) + \gamma} \mathbb{I}_h^k(s, a)$

    # Bonus computation

    Set $\hat{B}_{H+1}^k(s, a) = 0$ for every $(s, a) \in \mathcal{S} \times \mathcal{A}$.

    **for** $h = H, H - 1, \dots, 1$ **do**

        **for** $(s, a) \in \mathcal{S} \times \mathcal{A}$ **do**

        $\tilde{b}_h^k(s) = \sum_{a \in \mathcal{A}} \frac{3\gamma H \pi_h^k(a|s)}{\overline{\mu}_h^k(s) \pi_h^k(a|s) + \gamma}; \quad \overline{b}_h^k(s) = \sum_{a \in \mathcal{A}} \frac{H \pi_h^k(a|s)(\overline{\mu}_h^k(s,a) - \underline{\mu}_h^k(s,a))}{\overline{\mu}_h^k(s) \pi_h^k(a|s) + \gamma}.$

        $b_h^k(s) = \tilde{b}_h^k(s) + \overline{b}_h^k(s).$

        $B_h^k(s, a) = b_h^k(s) + \max_{p' \in \mathcal{P}_h^k(s,a)} \sum_{s' \in \mathcal{S}, a' \in \mathcal{A}} p_h'(s' \mid s, a) \pi_{h+1}^k(a' \mid s') \hat{B}_{h+1}^k(s', a').$

        **end for**

    **end for**

    # Policy Improvement

    Define the policy $\pi^{k+1}$ for every $(s, a, h) \in \mathcal{S} \times \mathcal{A} \times [H]$ by:

$$\pi_h^{k+1}(a \mid s) = \frac{\pi_h^k(a \mid s) \exp\left(-\eta(\hat{U}_h^k(s, a) - B_h^k(s, a))\right)}{\sum_{a' \in \mathcal{A}} \pi_h^k(a' \mid s) \exp\left(-\eta(\hat{U}_h^k(s, a') - B_h^k(s, a'))\right)}.$$

**end for**

---

**Theorem C.1** (Restatement of Theorem 5.1). *Under unknown dynamics, running Algorithm 3 with $\eta = \frac{\sqrt{\iota}}{H\sqrt{SAK} + H^2\sqrt{K}}$ and $\gamma = 2\eta H$, guarantees with probability $1 - \delta$,*

$$R_K \lesssim H^3 S\sqrt{AK}\iota + H^4 S^3 A\iota^2.$$

*Proof.* Using Equation (2), we can break the regret of the algorithm as,

$$
R_K = \sum_{k=1}^{K} \sum_{h,s} \mu_h^\star(s) \left\langle \pi_h^k(\cdot \mid s) - \pi_h^\star(\cdot \mid s), U_h^k(s, \cdot) \right\rangle
$$

$$
= \underbrace{\sum_{k=1}^{K} \sum_{h,s} \mu_h^\star(s) \left\langle \pi_h^k(\cdot \mid s), U_h^k(s, \cdot) - \hat{U}_h^k(s, \cdot) \right\rangle}_{\text{BIAS}_1} + \underbrace{\sum_{k=1}^{K} \sum_{h,s} \mu_h^\star(s) \left\langle \pi_h^\star(\cdot \mid s), \hat{U}_h^k(s, a) - U_h^k(s, \cdot) \right\rangle}_{\text{BIAS}_2}
$$

$$
+ \underbrace{\sum_{k=1}^{K} \sum_{h,s} \mu_h^\star(s) \left\langle \pi_h^k(\cdot \mid s) - \pi_h^\star(\cdot \mid s), \hat{U}_h^k(s, \cdot) - B_h^k(s, \cdot) \right\rangle}_{\text{REG}} + \underbrace{\sum_{k=1}^{K} \sum_{h,s} \mu_h^\star(s) \left\langle \pi_h^k(\cdot \mid s) - \pi_h^\star(\cdot \mid s), B_h^k(s, \cdot) \right\rangle}_{\text{BONUS}},
$$

□

Due to optimism of the estimator $\text{BIAS}_2 \leq \tilde{O}(H^2/\gamma)$ under the good event $G_3$. In Lemmas C.7, C.8 and C.10 we bound $\text{BIAS}_1$, REG and BONUS. Overall we get that,

$$
R_K \leq \underbrace{\frac{2}{3} \sum_{k=1}^{K} \sum_{h,s} \mu_h^\star(s) \tilde{b}_h^k(s) + \sum_{k=1}^{K} \sum_{h,s} \mu_h^\star(s) \bar{b}_h^k(s) + O\left(\frac{H^2}{\gamma}\iota\right)}_{\text{BAIS}_1} + \underbrace{O\left(\frac{H^2}{\gamma}\iota\right)}_{\text{BAIS}_2}
$$

$$
+ \underbrace{\frac{H \ln A}{\eta} + \frac{1}{3} \sum_{k,h,s} \mu_h^\star(s) \tilde{b}_h^k(s) + O\left(\eta H^5 K + \frac{H^2}{\gamma}\iota\right)}_{\text{REG}}
$$

$$
+ \underbrace{O\left(\gamma H^2 SAK + H^3 S\sqrt{AK}\iota + H^4 S^3 A\iota^2\right) - \sum_{k=1}^{K} \sum_{h,s} \mu_h^\star(s) b_h^k(s)}_{\text{BONUS}}
$$

$$
\leq O\left(\frac{H \ln A}{\eta} + \gamma H^2 SAK + \eta H^5 K + \frac{H^2}{\gamma}\iota + H^3 S\sqrt{AK}\iota + H^4 S^3 A\iota^2\right)
$$

Plugging $\eta$ and $\gamma$ completes the proof.

## C.1. Good event

We define the following good event $G = \bigcap_{i=1}^{5} G_i$ which holds with high probability (Lemma C.5):

$$G_1 = \left\{ \forall (k, s', s, a, h). \; \left| p_h(s' \mid s, a) - \bar{p}_h^k(s' \mid s, a) \right| \leq 4\sqrt{\frac{\bar{p}_h^k(s' \mid s, a) \log \frac{6HSAK}{\delta}}{\max\{n_h^k(s, a), 1\}}} + 10 \frac{\log \frac{6HSAK}{\delta}}{\max\{n_h^k(s, a), 1\}} \right\}$$

$$G_2 = \left\{ \sum_{h,s,a,k} |\bar{\mu}_h^k(s, a) - \underline{\mu}_h^k(s, a)| \leq O\left( \sqrt{H^4 S^2 AK \log \frac{6KHSA}{\delta}} + H^3 S^3 A \log^2 \frac{6KHSA}{\delta} \right) \right\}$$

$$G_3 = \left\{ \sum_{k=1}^{K} \sum_{h,s,a} \mu_h^\star(s) \pi_h^k(a \mid s) \left( \mathbb{E}_k\left[\hat{U}_h^k(s, a)\right] - \hat{U}_h^k(s, a) \right) \leq \frac{1}{3} \sum_{k=1}^{K} \sum_{h,s} \mu_h^\star(s) \tilde{b}_h^k(s) + \frac{H^2 \log \frac{6}{\delta}}{\gamma} \right\}$$

$$G_4 = \left\{ \sum_{k=1}^{K} \sum_{h,s,a} \frac{\mu_h^\star(s) \pi_h^k(a \mid s) \mathbb{I}\{s_h^k = s, a_h^k = a\}}{(\bar{\mu}_h^k(s, a) + \gamma)^2} - \frac{\mu_h^\star(s) \pi_h^k(a \mid s)}{\bar{\mu}_h^k(s, a) + \gamma} \leq \frac{H \ln \frac{6H}{\delta}}{2\gamma^2} \right\}$$

$$G_5 = \left\{ \sum_{k=1}^{K} \sum_{h,s} \mu_h^\star(s) \left\langle \pi_h^\star(\cdot \mid s), \hat{U}_h^k(s, \cdot) - U_h^k(s, \cdot) \right\rangle \leq \frac{H^2}{2\gamma} \ln \frac{6H}{\delta} \right\}$$

Under the good event the regret will be deterministically bounded.

Event $G_1$ holds with high probability by standard Bernstein inequality (see, e.g., Lemma 2 in Jin et al. (2020)). As a consequence of event $G_1$, $p \in \mathcal{P}^k$ for all $k$. In particular, $\bar{\mu}_h^k(s, a) \geq \mu_h^k(s, a)$ for all $k, h, s$ and $a$. $G_2$ Holds with high probability by a standard techniques by Jin et al. (2020) of summing the confidence radius on the trajectory. $G_3, G_4$ and $G_5$ follow similar techniques as in Luo et al. (2021), adapted to our case.

**Lemma C.2** (Event $G_3$). *With probability $1 - \delta$,*

$$\sum_{k=1}^{K} \sum_{h,s,a} \mu_h^\star(s) \pi_h^k(a \mid s) \left( \mathbb{E}_k\left[\hat{U}_h^k(s, a)\right] - \hat{U}_h^k(s, a) \right) \leq \frac{1}{3} \sum_{k=1}^{K} \sum_{h,s} \mu_h^\star(s) b_h^k(s) + \frac{H^2 \log \frac{1}{\delta}}{\gamma}$$

*Proof.* Similarly to Lemma B.2, let $Y_k = \sum_{h,s} \mu_h^\star(s) \left\langle \pi_h^k(\cdot \mid s), \hat{U}_h^k(s, \cdot) \right\rangle$. To bound $\sum_k \mathbb{E}_k[Y_k] - \sum_k Y_k$ we'll use a

form of Freedman's Inequality (Lemma E.2). For that we need to bound $\mathbb{E}_k[(Y_k - \mathbb{E}_k[Y_k])^2]$:

$$\mathbb{E}_k[(Y_k - \mathbb{E}_k[Y_k])^2] \leq \mathbb{E}_k[Y_k^2]$$

$$= \mathbb{E}_k\left[\left(\sum_{h,s,a} \mu_h^\star(s)\pi_h^k(a \mid s)\hat{U}_h^k(s,a)\right)^2\right]$$

$$\leq \mathbb{E}_k\left[\left(\sum_{h,s,a} \mu_h^\star(s)\pi_h^k(a \mid s)\right)\left(\sum_{h,s,a} \mu_h^\star(s)\pi_h^k(a \mid s)\left(\hat{U}_h^k(s,a)\right)^2\right)\right] \qquad \text{(Cauchy–Schwarz)}$$

$$= H\mathbb{E}_k\left[\sum_{h,s,a} \mu_h^\star(s)\pi_h^k(a \mid s)\left(\hat{U}_h^k(s,a)\right)^2\right]$$

$$= H\mathbb{E}_k\left[\sum_{h,s,a} \mu_h^\star(s)\pi_h^k(a \mid s)\frac{(L_{1:H}^k)^2}{(\overline{\mu}_h^k(s,a) + \gamma)^2}\mathbb{I}_h^k(s,a)\right]$$

$$\leq H^3\mathbb{E}_k\left[\sum_{h,s,a} \mu_h^\star(s)\pi_h^k(a \mid s)\frac{\mathbb{I}_h^k(s,a)}{(\overline{\mu}_h^k(s,a) + \gamma)^2}\right] \qquad (L_{1:H}^k \leq H)$$

$$= H^3 \sum_{h,s,a} \mu_h^\star(s)\pi_h^k(a \mid s)\frac{\mu_h^k(s,a)}{(\overline{\mu}_h^k(s,a) + \gamma)^2} \qquad (\mathbb{E}_k[\mathbb{I}_h^k(s,a)] = \mu_h^k(s,a))$$

$$\leq H^3 \sum_{h,s,a} \mu_h^\star(s)\frac{\pi_h^k(a \mid s)}{\overline{\mu}_h^k(s,a) + \gamma}. \qquad (\mu_h^k(s,a) \leq \overline{\mu}_h^k(s,a) \text{ under } G_1)$$

By Lemma E.2 with probability $1 - \delta$,

$$\sum_k \mathbb{E}_k[Y_k] - \sum_k Y_k \leq \alpha \sum_k \mathbb{E}_k[(Y_k - \mathbb{E}_k[Y_k])^2] + \frac{\log\frac{1}{\delta}}{\alpha}$$

where $\alpha \in (0, 1/R]$ for $R$ such that $|Y_k - \mathbb{E}_k[Y_k]| \leq R$ for any $k$. In our case, $|Y_k - \mathbb{E}_k[Y_k]| \leq H^2/\gamma$. And so,

$$\sum_k \mathbb{E}_k[Y_k] - \sum_k Y_k \leq \frac{1}{3}\sum_{k=1}^K \sum_{h,s} \mu_h^\star(s)\tilde{b}_h^k(s) + \frac{H^2 \log\frac{1}{\delta}}{\gamma}.$$

$\square$

**Lemma C.3** (Event $G_4$). *With probability $1 - \delta$,*

$$\sum_{k=1}^K \sum_{h,s,a} \frac{\mu_h^\star(s)\pi_h^k(a \mid s)\mathbb{I}\{s_h^k = s, a_h^k = a\}}{(\overline{\mu}_h^k(s,a) + \gamma)^2} - \frac{\mu_h^\star(s)\pi_h^k(a \mid s)}{\overline{\mu}_h^k(s,a) + \gamma} \leq \frac{H \ln\frac{H}{\delta}}{2\gamma^2}$$

*Proof.* Follows directly from Lemma E.4 with with $Z_h^k(s,a) = z_h^k(s,a) = \frac{\mu_h^\star(s)\pi_h^k(a|s)}{\overline{\mu}_h^k(s,a)+\gamma} \leq \frac{1}{\gamma}$ and $\tilde{\mu}_h^k(s,a) = \overline{\mu}_h^k(s,a)$. $\square$

**Lemma C.4** (Event $G_5$). *With probability $1 - \delta$,*

$$\sum_{k=1}^K \sum_{h,s} \mu_h^\star(s)\left\langle \pi_h^\star(\cdot \mid s), \hat{U}_h^k(s,\cdot) - U_h^k(s,\cdot)\right\rangle \leq \frac{H^2}{2\gamma} \ln\frac{H}{\delta}$$

*Proof.* Similar to Lemma C.4, we invoke Lemma E.4 with $Z_h^k(s,a) = \mu_h^\star(s)\pi_h^k(a,s)[\mathbb{I}_h^k(s,a)L_{1:H}^k + (1 - \mathbb{I}_h^k(s,a))U_h^k(s,a)] \le H$, $z_h^k(s,a) = \mathbb{E}_k[Z_h^k(s,a)] = \mu_h^\star(s)\pi_h^k(a,s)U_h^k(s,a)$ and $\tilde{\mu}_h^k(s,a) = \overline{\mu}_h^k(s,a)$, we get,

$$\sum_{k=1}^K \sum_{h,s} \mu_h^\star(s) \left\langle \pi_h^\star(\cdot \mid s), \hat{U}_h^k(s,\cdot) - U_h^k(s,\cdot) \right\rangle$$

$$= \sum_{k=1}^K \sum_{h,s,a} \frac{\mu_h^\star(s)\pi_h^k(a,s)\mathbb{I}_h^k(s,a)L_{1:H}^k}{\overline{\mu}_h^k(s,a) + \gamma} - \sum_{k=1}^K \sum_{h,s,a} \mu_h^\star(s)\pi_h^k(a,s)U_h^k(s,a) \le \frac{H^2}{2\gamma} \ln \frac{H}{\delta}$$

$\square$

**Lemma C.5.** *Under Algorithm 3, the good event $G = \bigcap_{i=1}^5 G_i$ holds with probability of at least $1 - \delta$.*

*Proof.* The proof directly follows from invoking Lemmas C.2 to C.4, E.6 and E.7 with $\delta/5$ and taking the union bound. $\square$

### C.2. Bound on BIAS$_1$

**Lemma C.6.** *For any $h, s, a$ and $k$,*

$$\mathbb{E}\left[\hat{U}_h^k(s,a) \mid \pi^k\right] = \frac{\mu_h^k(s,a)}{\overline{\mu}_h^k(s,a) + \gamma} U_h^k(s,a)$$

*Proof.* By definition $\Pr(\mathbb{I}_h^k(s,a) = 1 \mid \pi^k) = \mu_h^k(s,a)$. Using the law of total expectation and the fact that $\hat{U}_h^k(s,a) = 0$ whenever $\mathbb{I}_h^k(s,a) = 0$ we get,

$$\mathbb{E}_k\left[\hat{U}_h^k(s,a)\right] = \mathbb{E}_k\left[\hat{U}_h^k(s,a) \mid \mathbb{I}_h^k(s,a) = 1\right] \cdot \mu_h^k(s,a) + \underbrace{\mathbb{E}_k\left[\hat{U}_h^k(s,a) \mid \mathbb{I}_h^k(s,a) = 0\right]}_{=0}(1 - \mu_h^k(s,a))$$

$$= \mathbb{E}_k\left[\frac{\sum_{h'=1}^H \ell_{h'}^k(s_{h'}^k, a_{h'}^k)}{\overline{\mu}_h^k(s,a) + \gamma} \mid \mathbb{I}_h^k(s,a) = 1\right] \cdot \mu_h^k(s,a)$$

$$= \frac{\mu_h^k(s,a)}{\overline{\mu}_h^k(s,a) + \gamma} \mathbb{E}_k\left[\sum_{h'=h}^H \ell_{h'}^k(s_{h'}^k, a_{h'}^k) \mid s_h^k = s, a_h^k = a\right]$$

$$= \frac{\mu_h^k(s,a)}{\overline{\mu}_h^k(s,a) + \gamma} U_h^k(s,a).$$

$\square$

**Lemma C.7.** *Under the good event,*

$$\text{BIAS}_1 \le \frac{2}{3} \sum_{k=1}^K \sum_{h,s} \mu_h^\star(s)\tilde{b}_h^k(s) + \sum_{k=1}^K \sum_{h,s} \mu_h^\star(s)\overline{b}_h^k(s) + O\left(\frac{H^2}{\gamma}\iota\right).$$

*Proof.* Let $Y_k = \sum_{h,s} \mu_h^\star(s) \left\langle \pi_h^k(\cdot \mid s), \hat{U}_h^k(s,\cdot) \right\rangle$. It holds that,

$$\text{BIAS}_1 = \sum_{k=1}^K \sum_{h,s} \mu_h^\star(s) \left\langle \pi_h^k(\cdot \mid s), U_h^k(s,\cdot) \right\rangle - \sum_{k=1}^K \mathbb{E}_k[Y_k] + \sum_{k=1}^K \mathbb{E}_k[Y_k] - \sum_{k=1}^K Y_k.$$

Under the good event, $G_3$, it holds that

$$\sum_{k=1}^K \mathbb{E}_k[Y_k] - \sum_{k=1}^K Y_k \le \frac{1}{3}\sum_{k=1}^K \sum_{h,s} \mu_h^\star(s)\tilde{b}_h^k(s) + \frac{H^2}{\gamma} \ln \frac{10}{\delta}.$$

Using Lemma C.6, and the fact that under the good event $\underline{\mu}_h^k(s,a) \leq \mu_h^k(s,a)$,

$$\sum_{k=1}^{K}\sum_{h,s}\mu_h^\star(s)\left\langle \pi_h^k(\cdot \mid s), U_h^k(s,\cdot)\right\rangle - \sum_{k=1}^{K}\mathbb{E}_k[Y_k] = \sum_{k=1}^{K}\sum_{h,s,a}\mu_h^\star(s)\pi_h^k(a \mid s)U_h^k(s,a)\left(1 - \frac{\mu_h^k(s,a)}{\overline{\mu}_h^k(s,a)+\gamma}\right)$$

$$= \sum_{k=1}^{K}\sum_{h,s,a}\mu_h^\star(s)\pi_h^k(a \mid s)U_h^k(s,a)\frac{\gamma + \overline{\mu}_h^k(s,a) - \mu_h^k(s,a)}{\overline{\mu}_h^k(s,a)+\gamma}$$

$$\leq \sum_{k=1}^{K}\sum_{h,s,a}\mu_h^\star(s)\pi_h^k(a \mid s)U_h^k(s,a)\frac{\gamma}{\overline{\mu}_h^k(s,a)+\gamma}$$

$$+ \sum_{k=1}^{K}\sum_{h,s,a}\mu_h^\star(s)\pi_h^k(a \mid s)U_h^k(s,a)\frac{\overline{\mu}_h^k(s,a) - \underline{\mu}_h^k(s,a)}{\overline{\mu}_h^k(s,a)+\gamma}$$

(under the good event $\underline{\mu}_h^k(s,a) \leq \mu_h^k(s,a)$)

The first term above is bounded by,

$$\sum_{k=1}^{K}\sum_{h,s,a}\mu_h^\star(s)\pi_h^k(a \mid s)U_h^k(s,a)\frac{\gamma}{\overline{\mu}_h^k(s,a)+\gamma} \leq \sum_{k=1}^{K}\sum_{h,s,a}\mu_h^\star(s)\pi_h^k(a \mid s)\frac{H\gamma}{\overline{\mu}_h^k(s,a)+\gamma}$$

$$= \frac{1}{3}\sum_{k=1}^{K}\sum_{h,s,a}\mu_h^\star(s)\tilde{b}_h^k(s)$$

The second term is bounded by,

$$\sum_{k=1}^{K}\sum_{h,s,a}\mu_h^\star(s)\pi_h^k(a \mid s)U_h^k(s,a)\frac{\overline{\mu}_h^k(s,a) - \underline{\mu}_h^k(s,a)}{\overline{\mu}_h^k(s,a)+\gamma} \leq \sum_{k=1}^{K}\sum_{h,s,a}\mu_h^\star(s)\pi_h^k(a \mid s)H\frac{\overline{\mu}_h^k(s,a) - \underline{\mu}_h^k(s,a)}{\overline{\mu}_h^k(s,a)+\gamma}$$

$$= \sum_{k=1}^{K}\sum_{h,s,a}\mu_h^\star(s)\overline{b}_h^k(s)$$

$\square$

## C.3. Bound on REG

**Lemma C.8.** *For $\eta \leq \frac{\gamma}{2H}$, under the good event of Lemma B.3,*

$$\text{REG} \leq \frac{H\ln A}{\eta} + 16\eta H^5 K + \frac{1}{3}\sum_{k,h,s}\mu_h^\star(s)b_h^k(s) + O\left(\frac{H^2\iota}{\gamma}\right)$$

*Proof.* Using standard entropy regularized OMD guarantee (Lemma E.5),

$$\text{REG} \leq \frac{H\ln A}{\eta} + 2\eta\sum_{k=1}^{K}\sum_{h,s,a}\mu_h^\star(s)\pi_h^k(a \mid s)\left(\hat{U}_h^k(s,a) - \hat{B}_h^k(s,a)\right)^2$$

$$\leq \frac{H\ln A}{\eta} + 2\eta\sum_{k=1}^{K}\sum_{h,s,a}\mu_h^\star(s)\pi_h^k(a \mid s)\hat{U}_h^k(s,a)^2 + 16\eta H^5 K,$$

where in the second inequality we use the fact that $b_h^k(s) \leq 4H$ and thus, $\hat{B}_h^k(s,a) \leq 4H^2$. For the middle term,

$$
2\eta \sum_{k,h,s,a} \mu_h^\star(s)\pi_h^k(a \mid s)\hat{U}_h^k(s,a)^2 \leq 2\eta \sum_{k,h,s,a} \mu_h^\star(s)\pi_h^k(a \mid s)\frac{H^2\mathbb{I}\{s_h^k = s, a_h^k = a\}}{(\overline{\mu}_h^k(s,a) + \gamma)^2}
$$

$$
\leq 2\eta H^2 \sum_{k,h,s,a} \frac{\mu_h^\star(s)\pi_h^k(a \mid s)}{\overline{\mu}_h^k(s,a) + \gamma} + O\left(\eta\frac{H^3\iota}{\gamma^2}\right)
$$

$$
= \frac{2\eta}{3\gamma}H \sum_{k,h,s} \mu_h^\star(s)\tilde{b}_h^k(s) + O\left(\eta\frac{H^3\iota}{\gamma^2}\right)
$$

$$
\leq \frac{1}{3} \sum_{k,h,s} \mu_h^\star(s)\tilde{b}_h^k(s) + O\left(\frac{H^2\iota}{\gamma}\right).
$$

The second inequality above is under the good event $G_4$, and the last inequality is since $\eta \leq \frac{H}{2\gamma}$. $\qquad\square$

## C.4. Bound on BONUS

**Lemma C.9.** *Under the good event,*

$$
\text{BONUS} \leq \sum_{k,h,s} \overline{\mu}_h^k(s)b_h^k(s) - \sum_{k,h,s} \mu_h^\star(s)b_h^k(s).
$$

*Proof.* Let $\hat{p}^k$ be the transition function chosen by the algorithm when calculating $\hat{B}^k$. It holds that

$$
\sum_{h,s}\mu_h^*(s)\left\langle \pi_h^k(\cdot \mid s) - \pi_h^*(\cdot \mid s), \hat{B}_h^k(s,\cdot)\right\rangle =
$$

$$
= \sum_{h,s,a} \mu_h^*(s)\pi_h^k(a \mid s)\hat{B}_h^k(s,a) - \sum_{h,s,a} \mu_h^*(s)\pi_h^*(a \mid s)\hat{B}_h^k(s,a)
$$

$$
= \sum_{h,s,a} \mu_h^*(s)\pi_h^k(a \mid s)\hat{B}_h^k(s,a) - \sum_{h,s,a} \mu_h^*(s)\pi_h^*(a \mid s)\left(b_h^k(s) + \sum_{s',a'}\hat{p}_h^k(s' \mid s,a)\pi_{h+1}^k(a' \mid s')\hat{B}_{h+1}^k(s',a')\right)
$$

$$
\leq \sum_{h,s,a} \mu_h^*(s)\pi_h^k(a \mid s)\hat{B}_h^k(s,a) - \sum_{h,s,a} \mu_h^*(s)\pi_h^*(a \mid s)\left(b_h^k(s) + \sum_{s',a'}p_h(s' \mid s,a)\pi_{h+1}^k(a' \mid s')\hat{B}_{h+1}^k(s',a')\right)
$$

$$
= \underbrace{\sum_{h,s,a} \mu_h^*(s)\pi_h^k(a \mid s)\hat{B}_h^k(s,a) - \sum_{h,s,a} \mu_{h+1}^*(s)\pi_{h+1}^k(a \mid s)\hat{B}_{h+1}^k(s,a)}_{(i)} - \sum_{h,s}\mu_h^*(s)b_h^k(s) \qquad (*)
$$

where the inequality is since $p \in \mathcal{P}^k$ under event $G_1$, and $\hat{p}^k$ maximizes the term in the parentheses. $(*)$ uses $\sum_{s,a} \mu_h^\star(s)\pi_h^\star(a \mid s)p_h(s' \mid s,a) = \mu_{h+1}^\star(s')$ and then change the variables $s'$ and $a'$ to $s$ and $a$. $(i)$ is a telescopic sum, and recall that $\hat{B}_{H+1}^k \equiv 0$. Thus,

$$
(i) = \sum_{s,a} \mu_1^*(s)\pi_1^k(a \mid s)\hat{B}_1^k(s,a) = \sum_a \pi_1^k(a \mid s_{init})\hat{B}_1^k(s_{init},a) \qquad (\mu_1^*(s) = \mathbb{I}\{s = s_{init}\})
$$

$$
= V_1^{\pi^k,\hat{p}^k}(s_{init})
$$

$$
= \sum_{h,s} \mu_h^{\pi^k,\hat{p}^k}(s)b_h^k(s)
$$

$$
\leq \sum_{h,s} \overline{\mu}_h^k(s)b_h^k(s),
$$

where the inequality is since $\hat{p}^k \in \mathcal{P}^k$, and $\overline{\mu}_h^k(s)$ is the maximal occupancy with respect to transitions in $\mathcal{P}^k$. Plugging $(i)$ back in the last display completes the proof. $\qquad\square$

**Lemma C.10.** *Under the good event,*

$$\text{BONUS} \leq O\left(\gamma H^2 SAK + H^3 S\sqrt{AK}\iota + H^4 S^3 A\iota^2\right) - \sum_{k,h,s} \mu_h^\star(s) b_h^k(s).$$

*Proof.* From lemma Lemma C.9,

$$\text{BONUS} \leq \sum_{k,h,s} \overline{\mu}_h^k(s) \tilde{b}_h^k(s) + \sum_{k,h,s} \overline{\mu}_h^k(s) \overline{b}_h^k(s) - \sum_{k,h,s} \mu_h^\star(s) b_h^k(s).$$

The first term is bounded by,

$$\sum_{k=1}^{K} \sum_{h,s} \overline{\mu}_h^k(s) \tilde{b}_h^k(s) = 3\gamma H \sum_{k=1}^{K} \sum_{h,s,a} \frac{\overline{\mu}_h^k(s) \pi_h^k(a \mid s)}{\overline{\mu}_h^k(s) \pi_h^k(a \mid s) + \gamma} \leq 3\gamma H^2 SAK.$$

For the second term we use the good event $G_2$,

$$
\begin{aligned}
\sum_{k,h,s} \overline{\mu}_h^k(s) \overline{b}_h^k(s) &= H \sum_{k,h,s} \overline{\mu}_h^k(s,a) \frac{\overline{\mu}_h^k(s,a) - \underline{\mu}_h^k(s,a)}{\overline{\mu}_h^k(s,a) + \gamma} \\
&\leq H \sum_{k,h,s} |\overline{\mu}_h^k(s,a) - \underline{\mu}_h^k(s,a)| \\
&\leq H^3 S\sqrt{AK}\iota + H^4 S^3 A\iota^2,
\end{aligned}
$$

where the last inequality is by Lemma E.7. $\qquad\square$

# D. Lower Bound

**Theorem D.1** (Restatement of Theorem 6.2). *Assume that $H, S, A \geq 2$ and $K \geq 2S$. Any learning algorithm for the online MDPs with known dynamics and aggregate bandit feedback problem must incur at least $\Omega(H^2\sqrt{SAK})$ expected regret in the worst case.*

*Proof.* Consider an MDP with $S$ states: $s_1, s_2, \ldots, s_S$ where $s_1$ is the initial state. The idea is to encode a hard multitask bandit problem with $H - 1$ tasks in each of the states. The agent starts in the initial state $s_1$ where the loss is $0$ and any action transitions to each of the states $s_1, \ldots, s_S$ with probability $1/S$. I.e., $p_1(s' \mid a, s_1) = 1/S$ for any $s'$ and $a$. From time $h \geq 2$ the agent stays at the same state, $p_h(s' \mid s, a) = \mathbb{I}\{s' = s\}$. Each state $s_i$ encodes a hard multitask bandit problem with $H - 1$ tasks. That is, the losses are generated (independently for each state) from the (randomized) instance that attains the lower bound of Lemma 6.1.

Denote by $T_i$ the number of times we transition to $s_i$. From Lemma 6.1 the expected regret from visits at $s_i$ is at least $\Omega(\mathbb{E}[H^2\sqrt{AT_i}])$. In total, we have a lower bound on the regret of

$$\Omega\left(\mathbb{E}\left[H^2\sum_{i=1}^{S}\sqrt{SAT_i}\right]\right) = \Omega\left(H^2 S\sqrt{A}\mathbb{E}[\sqrt{X}]\right),$$

for $X \sim Bin(n = K, p = 1/S)$ since each $T_i$ is a binomial random variable with parameters $K$ and $1/S$. By Lemma E.8, we have $\mathbb{E}[\sqrt{X}] \geq \tilde{\Omega}(\sqrt{np}) = \tilde{\Omega}(\sqrt{K/S})$ for $K \geq 2S$ which proves the lower bound $\Omega(H^2\sqrt{SAK})$. $\square$

# E. Auxiliary Lemmas

**Lemma E.1** (Azuma–Hoeffding inequality). *Let $\{X_t\}_{t\geq 1}$ be a real valued martingale difference sequence adapted to a filtration $\mathcal{F}_1 \subseteq \mathcal{F}_2 \subseteq \ldots$ (i.e., $\mathbb{E}[X_t \mid \mathcal{F}_t] = 0$). If $|X_t| \leq R$ a.s. then with probability at least $1 - \delta$,*

$$\sum_{t=1}^{T} X_t \leq R\sqrt{T \ln \frac{1}{\delta}}.$$

**Lemma E.2** (A special form of Freedman's Inequality, Theorem 1 of Beygelzimer et al. (2011)). *Let $\{X_t\}_{t\geq 1}$ be a real valued martingale difference sequence adapted to a filtration $\mathcal{F}_1 \subseteq \mathcal{F}_2 \subseteq \ldots$ (i.e., $\mathbb{E}[X_t \mid \mathcal{F}_t] = 0$). If $|X_t| \leq R$ a.s. then for any $\alpha \in (0, 1/R), T \in \mathbb{N}$ it holds with probability at least $1 - \delta$,*

$$\sum_{t=1}^{T} X_t \leq \alpha \sum_{t=1}^{T} \mathbb{E}[X_t^2 \mid \mathcal{F}_t] + \frac{\log(1/\delta)}{\alpha}.$$

**Lemma E.3** (Consequence of Freedman's Inequality, e.g., Lemma E.2 in (Cohen et al., 2021a)). *Let $\{X_t\}_{t\geq 1}$ be a sequence of random variables, supported in $[0, R]$, and adapted to a filtration $\mathcal{F}_1 \subseteq \mathcal{F}_2 \subseteq \ldots$. For any $T$, with probability $1 - \delta$,*

$$\sum_{t=1}^{T} X_t \leq 2\mathbb{E}[X_t \mid \mathcal{F}_t] + 4R \log \frac{1}{\delta}.$$

**Lemma E.4** (Lemma A.2 of Luo et al. (2021)). *Given a filtration $\mathcal{F}_0 \subseteq \mathcal{F}_1 \subseteq \ldots$, let $z_h^k(s,a) \in [0, R]$ and $\tilde{\mu}_h^k(s,a) \in [0,1]$ be sequences of $\mathcal{F}_k$-measurable functions. If $Z_h^k(s,a) \in [0, R]$ is a sequence of random variables such that $\mathbb{E}[Z_h^k(s,a) \mid \mathcal{F}_k] = z_h^k(s,a)$ then with probability $1 - \delta$,*

$$\sum_{k=1}^{K} \sum_{h,s,a} \frac{\mathbb{I}\{s_h^k = s, a_h^k = s\} Z_h^k(s,a)}{\tilde{\mu}_h^k(s,a) + \gamma} - \sum_{k=1}^{K} \sum_{h,s,a} \frac{\mu_h^k(s,a) z_h^k(s,a)}{\tilde{\mu}_h^k(s,a)} \leq \frac{RH}{2\gamma} \ln \frac{H}{\delta}$$

**Lemma E.5** (Standard entropy regularized OMD guarantee, see e.g., (Hazan et al., 2016)). *Let $\eta > 0$, and an arbitrary sequence $\{g_k\}_{k=1}^{K}$ such that for all $k \in [K]$, $a \in [d]$, $g_k \in \mathbb{R}^d$ and $\eta g_k(a) \geq -1$. Let $x_k \in \Delta_d$ be a sequence of vectors such that for all $a$, $x_1(a) = 1/n$, for all $k \in [K]$, $a \in [d]$,*

$$x_{k+1}(a) = \frac{x_k(a) e^{-\eta g_k(a)}}{\sum_{a' \in [n]} x_k(a') e^{-\eta g_k(a')}}.$$

*Then for any $x^\star \in \Delta_d$,*

$$\sum_{k=1}^{K} \langle g_k, x_k - x \rangle \leq \frac{\log d}{\eta} + \eta \sum_{k=1}^{K} \sum_{i=1}^{d} x_k(i) g_k(i)^2.$$

**Lemma E.6** (Lemma 2 in Jin et al. (2020)). *With probability $1 - \delta$, for all $(k, s', s, a, h)$,*

$$\left| p_h(s' \mid s, a) - \bar{p}_h^k(s' \mid s, a) \right| \leq 4\sqrt{\frac{\bar{p}_h^k(s' \mid s, a) \log \frac{HSAK}{\delta}}{\max\{n_h^k(s,a), 1\}}} + 10 \frac{\log \frac{HSAK}{\delta}}{\max\{n_h^k(s,a), 1\}}$$

**Lemma E.7** (Lemma 4 in Jin et al. (2020)). *With probability $1 - \delta$,*

$$\sum_{h,s,a,k} |\bar{\mu}_h^k(s,a) - \underline{\mu}_h^k(s,a)| \leq O\left(\sqrt{H^4 S^2 AK \log \frac{KHSA}{\delta}} + H^3 S^3 A \log^2 \frac{KHSA}{\delta}\right)$$

**Lemma E.8.** *Let $X \sim Bin(n, p)$ and assume that $n \geq \frac{2}{p}$. Then, $\mathbb{E}[\sqrt{X}] \geq \frac{1}{4}\sqrt{np}$.*

*Proof.* By Markov inequality we have:

$$\mathbb{E}[\sqrt{X}] \geq \frac{\sqrt{np}}{2} \Pr\left[\sqrt{X} \geq \frac{\sqrt{np}}{2}\right] = \frac{\sqrt{np}}{2} \Pr\left[X \geq \frac{np}{4}\right] = \frac{\sqrt{np}}{2}\left(1 - \Pr\left[X < \frac{np}{4}\right]\right).$$

Thus, it suffices to show that $\Pr\left[X < \frac{np}{4}\right] \leq 1/2$ which follows immediately from Multiplicative Chernoff bound and the assumption that $n \geq 2/p$. $\qquad\square$

