# OpenReview forum: "Near-optimal Regret Using Policy Optimization in Online MDPs with Aggregate Bandit Feedback"
_ICML.cc/2025/Conference — ICML 2025 poster_

### Official Review · Reviewer_JehC · 2025-02-20

**Overall Recommendation:** 2

**Summary:**

This paper considers the problem of learning Adversarial (Tabular) MDPs with aggregated bandit feedback, which means only the total loss (instead of per-round losses) incurred in an episode is revealed. Using PO w.r.t. newly-proposed U-functions on each state,
* with known transitions, an $\tilde{\mathcal O}(H^2 \sqrt{SAK})$-regret algorithm is proposed, matching the lower bound up to logarithmic factors;
* without transition information, an $\tilde{\mathcal O}(H^3 S \sqrt{AK})$ upper bound is derived; and
* a lower bound of $\Omega(H^2 \sqrt{SAK})$ is given via the reduction to multi-task bandit problems.

**Claims And Evidence:**

Claims are supported by rigorous proofs.

**Essential References Not Discussed:**

The literature review is pretty complete, except for a tiny bit regarding the removal of dilated bonuses (see below).

**Experimental Designs Or Analyses:**

N/A

**Methods And Evaluation Criteria:**

The notion of regret is aligned with previous works in the literature, and the comparison with them is fair.

**Other Comments Or Suggestions:**

The removal of dilated bonuses may sound hand-waving by only saying "$B_h^k (s,a)$ is computed using standard Bellman equations, which is simpler and more intuitive than the dilated version of Luo et al. (2021)." It'd be good to refer the readers to (Lancewicki et al., ICML'23) and roughly explain it's because this specific $b_h^k(s,a)=\mathcal O(1)$, which means the extra term $\eta\sum_k B_k^2$ in `Reg` can be directly controlled as $\mathcal O(\eta K)=\mathcal O(\sqrt K)$ instead of the original approach $\eta \sum_k B_k^2 = \mathcal O(\sum_k B_k)$.

* Tal Lancewicki, Aviv Rosenberg, Dmitry Sotnikov. Delay-Adapted Policy Optimization and Improved Regret for Adversarial MDP with Delayed Bandit Feedback. ICML 2023.

**Other Strengths And Weaknesses:**

The results are good to have given the recent interest on MDPs with aggreggated bandit feedback in the literature. The observation of the U-function and the corresponding version of performance difference lemma is cool, which avoids the reduction to linear bandits by Cohen et al. (2021b). However, besides these, it looks like the remaining components like implicit exploration, confidence sets on transitions and occupancy measures, and PO analysis for AMDPs, are more-or-less standard. Therefore I feel it's really a borderline result and can be much more interesting if the authors can utilize more properties of the U-functions (unfortunately, as the authors discuss in the Conclusion part, the structure of U-functions is fragile when the state/action space become infinite).

**Questions For Authors:**

Aside from the innovation of U-function and corresponding performance difference lemma, are there other technical innovations that might be independent interest? As I mentioned, I feel the U-functions are nice and elegant, but are really restricted to the specific setup of aggregated bandit feedback (and do not generalize to function approximations since I am not sure whether it makes sense to assume the U-functions are linear in some $\phi(s,a)$'s, which is an analog of the linear-Q assumption in the literature).

If there're any technical contribution that I missed, I'd be happy to re-evaluate this paper.

**Relation To Broader Scientific Literature:**

I am sorry to say this, but as far as I can see, the main contributions seem to be really restricted to the specific setup of aggregated bandit feedback (detailed below).

**Theoretical Claims:**

I read the statements of all the lemmas and checked several main steps in the proof.

---

> ### Author Rebuttal · Authors · 2025-03-30
>
> Thank you for your constructive review. Below is our response to your comments and questions.
>
> > *Aside from the innovation of U-function and corresponding performance difference lemma, are there other technical innovations that might be independent interest?*
>
> Major contributions of our work:
> The key novelty lies in the introduction of the U-function, which is easy to estimate under aggregate bandit feedback, and the regret decomposition with respect to it. This contrasts with previous approaches that directly tried to estimate the loss function, inducing several challenges under aggregate bandit feedback such as controlling the estimation bias—this results in a relatively complex algorithm and technical analysis. We believe that the U-function is an interesting concept that elegantly circumvents previous technical challenges and results in an intuitive and simple technical analysis. We believe that this new concept could be insightful for future research on RL with aggregate bandit feedback.
>
> > *The removal of dilated bonuses may sound hand-waving... It'd be good to refer the readers to (Lancewicki et al., ICML'23)...*
>
> Thank you for your suggestion - we will add a reference to Lancewicki et al. (2023) and discuss the technical differences from Luo et al. (2021).

---

> > ### Comment · Reviewer_JehC · 2025-04-01
> >
> > Thank you for your response. I agree that the definition of U-function itself is interesting and overcomes disadvantages of previous approaches, making the aggregated bandit feedback as easy (or as hard) as the standard bandit feedback by enabling various standard tabular AMDP techniques to be applicable. Nevertheless, I am still a bit disappointed about the limitation of U-functions to tabular cases and the lack of new techniques that can be of broader interest. Therefore, I still maintain a borderline assessment.

---

### Official Review · Reviewer_actv · 2025-03-07

**Overall Recommendation:** 5

**Summary:**

This paper studies regret minimization in tabular MDPs with adversarial losses, fixed transition kernel, and aggregate/trajectory/full- bandit feedback, meaning that at the end of each episode, the algorithm only receives the total loss among all the $H$ visited state-action pairs (i.e., the entire trajectory), rather than the loss of each pair (aka semi-bandit feedback).

This paper provides algorithms to achieve $O(H^2\\sqrt{SAK})$ known-transition upper bound, and $O(H^3S\\sqrt{AK})$ unknown-transition upper bound (ignoring log factors). This paper also shows a matching lower bound for the known-transitioin case, hence characterizing the minimax regret. The unknown-transition upper bound is the same as the best known one in the semi-bandit setting.

Techinally, this is due to the Policy Optimization (PO) framework and a novel performance-difference-like regret decomposition w.r.t. the proposed "U function".

**Claims And Evidence:**

yes

**Essential References Not Discussed:**

no

**Experimental Designs Or Analyses:**

n/a

**Methods And Evaluation Criteria:**

yes

**Other Comments Or Suggestions:**

n/a

**Other Strengths And Weaknesses:**

Strengths:

The new regret decompostion is novel. This is also naturally used in the algorithm design (e.g., now we need to build loss estimate for the U function which is simple). I am impressed by this idea.

**Questions For Authors:**

The questions below are mainly for my curiosity and better understanding this paper:

1. Seems that the boundes are specific to the PO framework (and the regret decomoposition). Does this mean that it's unclear whether occupancy measure + FTRL/OMD can achieve the same bound (as they do not come with such regret decomposition)?

2. As long as we have this new regret decomposition, is the regret analysis pretty much the same as in Luo et al., 2021? If not, what are the key differences? (I don't view this as weakness, I just want to check my understanding).

3. With trajectory feedback, we do not even need the complicated dialated bonus as in Luo et al., 2021. Is it still because now the loss estimate is w.r.t U function (which is already doing some sort of global exploration/optimization)?

4. In the lower bound, why can not we utilize the unknown transition to prove a harder lower bound? What's the challenge?

5. Do the authors believe semi-bandit is statiscally as easy as full-bandit, or the upper bound in semi-bandit can be improved?

Thanks in advance for sharing any thoughts.

**Relation To Broader Scientific Literature:**

relevant audience include researchers study online learning, reinforcement learning, policy optimization

**Theoretical Claims:**

I check the proof sketchs in the main body. Look correct to me.

---

> ### Author Rebuttal · Authors · 2025-03-30
>
> Thank you for the positive review and the great questions. We would be happy to discuss any of the points below in the final version that the reviewer believes will improve the paper.
>
> **Q1:** Indeed, our bounds are specific to the PO framework. The only 'occupancy measure + FTRL/OMD' algorithm for aggregate bandit feedback (ABF) is by Cohen et al. (2021b), where it is still unclear how to improve them using the occupancy measure approach. Given the regret structure in terms of occupancy measures, it is natural to attempt estimating the loss function itself; however, handling estimation bias under unknown dynamics is quite challenging since the learner does not precisely know the played occupancy measure.
> Under known dynamics, it might be possible to achieve the optimal bound using 'occupancy measure + FTRL/OMD'. While the optimal bound for linear bandits over general convex sets is $d\sqrt{T}$, in some special cases, such as the simplex, it is possible to achieve tighter bounds. Therefore, there may be room to improve the bounds for the set of occupancy measures.
> This is an important and interesting open challenge for future research.
>
> **Q2:** Yes, the key novelty lies in the introduction of the U-function, which is easy to estimate under aggregate bandit feedback, and the regret decomposition with respect to it.
>
> **Q3:** No, in fact, one does not need the dilated component of Luo et al., 2021 (i.e., the $(1+\frac{1}{H})$ factor in the backup operator), even in the semi-bandit case. For prior evidence, see [1].
>
> [1] Lancewicki, T., Rosenberg, A., & Sotnikov, D. "Delay-adapted policy optimization and improved regret for adversarial MDP with delayed bandit feedback." ICML 2023
>
> **Q4:** That is a great question. The challenge in improving the lower bound under unknown dynamics is that ABF is not less informative about the dynamics, since the agent still observes the transitions. Thus, unlike in the semi-bandit case, under ABF it remains unclear whether unknown dynamics make the problem statistically harder.
>
> **Q5:** In the known dynamics case, full-bandit feedback is in fact statistically harder than semi-bandit feedback—the optimal bound in the semi-bandit case is $H \sqrt{SAK}$, and our lower bound for the full-bandit case is $H^2 \sqrt{SAK}$. As for the known dynamics case, it remains highly unclear, partly because the optimal bound in the semi-bandit case is also an open question. Under semi-bandit and unknown transitions, the best known upper bound is $H^2 S \sqrt{AK}$ (Jin et al., 2020), while the best known lower bound is $\Omega(H^{3/2} \sqrt{SAK})$ (Jin et al., 2018). If the latter is optimal under semi-bandit, then due to our lower bound, ABF is statistically harder. We also conjecture that the $H^2$ dependency in our lower bound is optimal and that the extra $H$ in our upper bound is an artifact of the PO method (as PO currently has this artifact in the semi-bandit case). If this conjecture is true, and if $H^2 S \sqrt{AK}$ is optimal under semi-bandit, then full-bandit would statistically be as easy as semi-bandit.

---

> > ### Comment · Reviewer_actv · 2025-04-07
> >
> > I've read the response from the authors and also the communication between the authors and other reviewers. While currently the results are limited to tabular MDP and it's non-trivial to extend them to (linear) function approx. scenario, I still appreciate the progress made on the understanding of statistical limits in learning tabular MDPs, so I'd like to continue to support this work and recommend acceptance.

---

### Official Review · Reviewer_y9Yt · 2025-03-10

**Overall Recommendation:** 4

**Summary:**

This paper studies online episodic MDPs with adversarial costs and aggregate bandit feedback. Under aggregate bandit feedback, the agent only observes the entire episode loss, making it less informative than the full information setting (the agent observes the full cost function), and the bandit/semi-bandit feedback setting (the agent observes the loss over the agent's trajectory). The paper explores both known and unknown transition function scenarios and introduces a policy optimization algorithm for each case. Using a new tool called the U-function, the proposed algorithms improve upon existing regret bounds in the literature for both settings.

**Claims And Evidence:**

All theoretical claims are followed by proofs in the appendix or in the main paper.

**Essential References Not Discussed:**

All essential references seems to be discussed.

**Experimental Designs Or Analyses:**

not applicable.

**Methods And Evaluation Criteria:**

As a theoretical paper, the proposed algorithms are well-suited to the problem. Since a key advantage of policy optimization algorithms is their closed-form solutions, implementation should be relatively straightforward. Therefore, despite the paper’s theoretical focus, it would be interesting to include some experimental demonstrations of the U-functions in practice.

**Other Comments Or Suggestions:**

No other comments.

**Other Strengths And Weaknesses:**

This is an interesting paper for the following reasons:

- The paper is well-written and organized.

- The concept of the U-functions is new and is a simple and elegant way of attacking the bandit aggregate feedback problem.

- While there are no additional technical novelties beyond the introduction of the U-function, demonstrating that the policy optimization algorithms from Luo et al. (2021), originally designed for bandit feedback, can be adapted, thanks to the U-functions, to the more challenging aggregate bandit feedback setting while maintaining comparable regret bounds is a significant contribution. It improves on previous results, and could open the way for new contributions on works of RL with trajectory feedback.

**Questions For Authors:**

I checked the proof for the case with known dynamics and it sounds correct, but I wonder if a similar result could also be obtained without the exploration bonuses (as here the dynamics are known), using similar results as the Appendix B.3 from Jin et al 2020.

*Jin et al., Learning adversarial markov decision processes with bandit feedback and unknwon transition, 2020.*

**Relation To Broader Scientific Literature:**

The paper improves the current state-of-the-art bound for online MDPs with adversarial losses and aggregate bandit feedback when the dynamics are known and in this case they are the first to establish the near-optimal regret bound, and in the unknown dynamics setting they improve the terms related to the horizon, the state space size, and the action space size when compared to the previous approaches from the literature.

The main idea of the paper, the U-functions, seem new. The U-function at a given state-action pair computes the expect cost on the entire trajectory given that the agent visit this state-action pair. On the other hand, once the U-function is introduced, the rest of the paper (the Policy Optimization algorithm, the dilated bonus equation, and the regret analysis) follow from the work of Luo et al. (2021). The difference being that instead of considering an estimation of the Q-function, the paper considers an estimation of the U-function.

**Theoretical Claims:**

Proofs for theoretical claims seem correct.

---

> ### Author Rebuttal · Authors · 2025-03-30
>
> Thank you for the positive and constructive review. Below is our response to your comments and questions.
>
> > *despite the paper’s theoretical focus, it would be interesting to include some experimental demonstrations of the U-functions in practice.*
>
> Our work is theoretically focused. Previous theoretical work has taken a more direct approach and faced challenges such as directly estimating the loss function and controlling the estimation bias in the regret bound. In our work, we introduce the concept of U-functions, which allows us to circumvent these challenges and handle aggregate bandit feedback much more elegantly. We hope this new concept will be insightful for future research, particularly for empirical studies and more practical applications. However, we believe that tabular MDPs would not be a good framework for demonstrating the practical applications of the U-function. Instead, an extensive empirical study that incorporates the U-function into a practical deep RL method, tested under a well-chosen set of benchmarks with large state spaces, would be more appropriate. We believe that this is an important area for future research and will include it in the future work section.
>
> > *…for the case with known dynamics… I wonder if a similar result could also be obtained without the exploration bonuses (as here the dynamics are known), using similar results as the Appendix B.3 from Jin et al 2020.*
>
> Jin et al. 2020 employs an occupancy-measure-based algorithm where, indeed, one does not need to incorporate bonuses in the known dynamics case. On the other hand, in the Policy Optimization method, it is not clear how to achieve $\sqrt{K}$ bounds without an exploration bonus. As mentioned in the paragraph starting at line 240, the purpose of the bonus in the known dynamics case is to address the distribution mismatch between $\mu^\star(s)$ and $\mu^k(s)$ that appears in the regret bound (e.g., line 294, right column). This distribution mismatch is rather orthogonal from the additional challenges in the unknown dynamics case, and stems mainly from the fact that the value difference lemma breaks the regret over states with weights $\mu^\star(s)$, and that we run the OMD locally in each state (unlike the method in Jin et al. 2020, which is the framework that Cohen et al. (2021b) builds upon).

---

### Official Review · Reviewer_3L6W · 2025-03-17

**Overall Recommendation:** 4

**Summary:**

The paper studies finite-horizon MDPs with adversarial losses under the aggregate bandit feedback model. In the known-dynamics case, the paper achieves the first optimal regret bound, while in the case of unknown dynamics it significantly improves the previous best known result.

**Claims And Evidence:**

Yes, the claims made in the paper are supported by clear and convincing evidence and proofs.

**Essential References Not Discussed:**

The paper covers adequately the related work.

**Experimental Designs Or Analyses:**

N.A.

**Methods And Evaluation Criteria:**

N/A. The paper is on the theory of reinforcement learning.

**Other Comments Or Suggestions:**

- (Line 326) Can the authors provide indicative works where the Bernstein sets are used ?
- Can the authors provide the time complexity of the proposed algorithms ?

**Other Strengths And Weaknesses:**

N/A.

**Questions For Authors:**

- In the unknown dynamics setting, how does Algorithm 2 compute $\mu$? What is the time complexity per episode for these steps?
- Have the authors tried a model-free approach? (that is, without estimating the transition probabilities)

**Relation To Broader Scientific Literature:**

I believe this paper is of great importance, as it solves an important problem of RL theory. More specifically, the paper deals with finite-horizon, tabular, adversarial MDPs under the relatively unexplored, but very interesting, full bandit setting.
- The paper proposes the first optimal algorithm under known dynamics, matching the lower bound (proved by the authors) and the regret of [1] (semi-bandit).
- The paper significantly improves the previous best regret upper bound of [2] under unknown dynamics.
- From a technical standpoint, the authors introduce a novel RL objective, namely the U-function, which facilitates the proposed algorithms and the analysis for the full bandit setting. Based on the U-function, using well-established techniques, the paper achieves good and interesting results in this problem.

[1] Zimin, A. and Neu, G. Online learning in episodic marko-
vian decision processes by relative entropy policy search.
In Advances in Neural Information Processing Systems
26: 27th Annual Conference on Neural Information Processing Systems 2013. Proceedings of a meeting held
December 5-8, 2013, Lake Tahoe, Nevada, United States,
pp. 1583–1591, 2013.

[2] Cohen, A., Kaplan, H., Koren, T., and Mansour, Y. Online
markov decision processes with aggregate bandit feedback. In Conference on Learning Theory, pp. 1301–1329.
PMLR, 2021b.

**Theoretical Claims:**

Yes. I checked many proofs of the paper, especially Appendix B. I have some minor points/questions to the authors:
- I think that the bonus b(s) should not have $\pi$ in the denominator (see, for example, the analysis in line 684).
- Line 695: The variance term must contain $\mathbb{E}[Y_k]$, since the martingale difference is $Y_k - \mathbb{E}[Y_k]$. However, I believe that such minor errors across the analysis do not change the complexity results of the paper.

---

> ### Author Rebuttal · Authors · 2025-03-30
>
> Thank you for the positive and constructive review. Below is our response to your comments and questions.
>
> > *I think that the bonus b(s) should not have π in the denominator (see, for example, the analysis in line 684)*
>
> Note that in line 684 the denominator has $\mu_h^k (s,a)$ which is by definition $\mu_h^k (s) \pi_h^k (a \mid s)$ (as in the bonus).
>
> > *Line 695: The variance term must contain E[Yk], since the martingale difference is Yk−E[Yk]. However, I believe that such minor errors across the analysis do not change the complexity results of the paper.*
>
> Thank you for spotting that. This is indeed not a proper use of Lemma E.2 but has a simple fix: for the martingale difference, $|Y_k - \mathbb{E}[Y_k]| \leq \max(Y_k, \mathbb{E}[Y_k]) \leq \frac{H^2}{\gamma}$, and the variance is bounded by the second moment: $\mathbb{E}[(|Y_k - \mathbb{E}[Y_k]|)^2] \leq \mathbb{E}[Y_k^2]$. We will correct this in the final version in line 695 and also in line 990 (which has the same minor error).
>
> > *(Line 326) Can the authors provide indicative works where the Bernstein sets are used?*
>
> Yes, in the context of Policy Optimization see for example Shani et al. (2020); Luo et al. (2021). But it is also widely used in other types of algorithms such as in occupancy-measure-based algorithms (Jin et al., 2020) and FTPL (Dai et al., 2020).
> We will reference those in line 326.
>
> Dai, Yan, Haipeng Luo, and Liyu Chen. "Follow-the-perturbed-leader for adversarial markov decision processes with bandit feedback."
>
> > *Can the authors provide the time complexity of the proposed algorithms?*
>
> For the known dynamics case, the state occupancy measure can be computed using the following recursive formula: $\mu_{h+1}^{\pi}(s') = \sum_{s,a} \mu_{h}^{\pi}(s) \pi_{h}(a \mid s) P(s' \mid s,a)$, so that the full occupancy measure can be computed in $O(H S^2 A)$. The bonus $B$ is calculated via backward dynamic programming also in $O(H S^2 A)$, and the policy computation is done in $O(H S A)$. Thus, the total complexity per iteration is $O(H S^2 A)$.
>
> For the unknown dynamics case, the upper/lower confidence occupancy measure computation can be done using Algorithm 3 in (Jin et al., 2020), with a complexity of $O(H S^2 A)$ per state. The same can be done for the computation of the bonus $\hat{B}$.
>
> We will include these details in the final version.
>
> > *In the unknown dynamics setting, how does Algorithm 2 compute μ? What is the time complexity per episode for these steps?*
>
> See above.
>
> > *Have the authors tried a model-free approach?*
>
> To the best of our knowledge, all existing regret minimization algorithms for MDPs with non-stochastic losses are model-based, even in the semi-bandit case. Achieving sub-linear regret under non-stochastic losses using a model-free algorithm is a very interesting future direction, already in the simpler context of semi-bandit feedback.

---

### Decision · Program_Chairs · 2025-05-01

**Decision:**

Accept (poster)

**Comment:**

This paper studies online episodic MDPs with adversarial costs and aggregate bandit feedback. The paper studies both known and unknown transition function scenarios and introduces a policy optimization algorithm for each case. Using a new tool called the U-function, in the known-dynamics case, the paper achieves the first optimal regret bound, while in the case of unknown dynamics it significantly improves the previous best known result.

Reviewers think the results are strong and there are novelties in the techniques. The AC agrees and thus recommends acceptance.